# A therapeutic combination of two small molecule toxin inhibitors provides broad preclinical efficacy against viper snakebite

Laura-Oana Albulescu[1], Chunfang Xie[2,3], Stuart Ainsworth[1], Jaffer Alsolaiss[1], Edouard Crittenden[1], Charlotte A. Dawson [1], Rowan Softley[1], Keirah E. Bartlett[1], Robert A. Harrison[1,4], Jeroen Kool [2,3] & Nicholas R. Casewell [1,4 ✉]

Snakebite is a medical emergency causing high mortality and morbidity in rural tropical communities that typically experience delayed access to unaffordable therapeutics. Viperid snakes are responsible for the majority of envenomings, but extensive interspecific variation in venom composition dictates that different antivenom treatments are used in different parts of the world, resulting in clinical and financial snakebite management challenges. Here, we show that a number of repurposed Phase 2-approved small molecules are capable of broadly neutralizing distinct viper venom bioactivities in vitro by inhibiting different enzymatic toxin families. Furthermore, using murine in vivo models of envenoming, we demonstrate that a single dose of a rationally-selected dual inhibitor combination consisting of marimastat and varespladib prevents murine lethality caused by venom from the most medically-important vipers of Africa, South Asia and Central America. Our findings support the translation of combinations of repurposed small molecule-based toxin inhibitors as broad-spectrum therapeutics for snakebite.

[1] Centre for Snakebite Research and Interventions, Liverpool School of Tropical Medicine, Pembroke Place, L3 5QA Liverpool, UK. [2] Division of BioAnalytical Chemistry, Amsterdam Institute of Molecular and Life Sciences, Vrije Universiteit Amsterdam, De Boelelaan 1085, 1081 HV Amsterdam, The Netherlands. [3] Centre for Analytical Sciences Amsterdam (CASA), 1098 XH Amsterdam, The Netherlands. [4] Centre for Drugs and Diagnostics, Liverpool School of Tropical Medicine, Pembroke Place, L3 5QA Liverpool, UK. ✉email: Nicholas.Casewell@lstmed.ac.uk

Snakebite is a neglected tropical disease (NTD) that causes extensive mortality (~138,000/annum) and morbidity (~400,000/annum) in the impoverished rural communities of sub-Saharan Africa, South and Southeast Asia, Oceania, and Central and South America[1]. Despite annual snakebite deaths equating to one-quarter of those that succumb to malaria[2], this NTD has long been overlooked by the global health community, resulting in little investment in snakebite management, the development of new therapeutics or improving speed of access to treatment. In 2017, snakebite was reclassified as a priority NTD by the World Health Organization (WHO) and, soon after, a global roadmap was published outlining the goal of halving snakebite-related deaths and disabilities by 2030[3]. Key tasks to achieve this goal include those relating to therapeutics, specifically the necessity to improve their safety, efficacy, affordability and accessibility to those in greatest need.

Snake venoms are complex mixtures of numerous proteins and peptides and extensive interspecific variation in venom composition poses major challenges for the development of generic (i.e., pancontinental) snakebite treatments[4,5]. Current therapies, known as antivenoms, consist of polyclonal immunoglobulins purified from the plasma/serum of large animals (e.g., equines, ovines) hyperimmunized with snake venoms. Because of the specificity of the resulting immunoglobulins towards the toxins present in the venoms used in manufacture, antivenoms typically have limited efficacy against envenoming by different snake species[6]. Consequently, distinct antivenom products are produced (>45 manufacturers worldwide) to treat envenoming by numerous snake species found in different parts of the world, resulting in a highly fragmented drug market, issues with affordability, and a lack of sustainability[7,8]. Other limitations with current antivenom include the following: (i) poor dose efficacy, as the majority (~80–90%) of their immunoglobulins do not bind venom toxins[1,9], (ii) high incidences of adverse reactions due to the administration of large doses of foreign immunoglobulins[10], (iii) the requirement for intravenous delivery in a healthcare facility, and (iv) reliance on cold chain transport and storage. In addition, many rural snakebite victims suffer major delays in accessing healthcare facilities following a bite, if they choose to attend at all, as evidenced by estimates suggesting that 75% of snakebite deaths occur outside of a hospital setting[11]. Cumulatively, these limitations identify an urgent and compelling need to develop cross-generically efficacious, stable and affordable, prehospital treatments as an effective means to considerably decrease snakebite mortality and morbidity[12,13].

Vipers represent a major group of medically important snakes that are widely distributed across the globe, ranging from the Americas to Africa and Asia, and are responsible for causing the majority of snake envenomings in these regions[14–16]. Treatments for systemic viper envenoming need to neutralize a number of major classes of hemotoxins, which are found in varying abundances across medically important snake species, and typically include the $Zn^{2+}$-dependent snake venom metalloproteinases (SVMPs), phospholipase $A_2$ (PLA$_2$s) and snake venom serine proteases (SVSPs)[17]. Collectively, these three enzymatic families typically comprise >60% of all toxins found in viper venom proteomes[5] and, in combination, are largely responsible for: (i) the destruction of local tissue, often resulting in necrosis, (ii) the degradation of the basement and cellular membranes resulting in extravasation, and (iii) the onset of coagulopathy via the activation and breakdown of clotting factors—with the latter two effects often culminating in life-threatening systemic hemorrhage[17–20].

Small molecule toxin inhibitors have received limited attention as potential alternatives to immunoglobulin-based snakebite therapies[12,21–25], although recent findings have suggested that a number of Phase-2 approved drugs may hold therapeutic

promise[23,26–28]. Perhaps the most notable of these is the PLA$_2$-inhibitor, varespladib, which has been widely explored for repurposing as a snakebite therapy, and has shown substantial promise in preclinical models against a number of elapid and viper venoms[22,26,27,29]. In addition, several SVMP-inhibitors have been demonstrated to be capable of abolishing venom-induced hemorrhage or dermonecrosis, including metal ion chelators[21,24,25,28] and peptidomimetic hydroxamate inhibitors[23,24,30]. We recently reported that 2,3-dimercapto-1-propanesulfonic acid (DMPS), a $Zn^{2+}$ chelator that is a licensed oral medicine used to treat heavy metal poisoning, was particularly effective in preclinically neutralizing both the local and systemic toxicity of $Zn^{2+}$-dependent SVMP-rich saw-scaled viper venoms (genus *Echis*)[28]. However, despite the promise of both varespladib and DMPS as orally delivered prehospital therapeutics for snakebite, both are likely to be somewhat restricted in terms of their efficacy, as each predominantly targets only one of the handful of major toxin families found in the venoms of medically important snakes.

To address this limitation, and cognisant of the complexity of snake venoms, herein we explored the potential of combinations of small molecule toxin inhibitors as new 'broad spectrum' snakebite therapeutics. Our goal—to rationally select and preclinically validate a therapeutic small molecule mixture capable of neutralizing distinct pathogenic toxins found in the venoms of geographically diverse, medically important, hemotoxic vipers—was achieved. Thus, we demonstrate, in a mouse model of envenoming, that a single dose of the SVMP-inhibitor marimastat, combined with the PLA$_2$-inhibitor varespladib, provides in vivo protection against the lethal effects of envenoming caused by the most medically important vipers of Africa, south Asia and Central America. Our findings suggest that combinations of small molecule toxin inhibitors are promising drug leads for the future development of generic prehospital therapies for treating hemotoxic snakebites.

## Results

**Venom SVMP activities are neutralized by peptidomimetic inhibitors and metal chelators.** SVMPs represent a major class of enzymatic toxins responsible for causing severe snakebite pathology, including hemorrhage, coagulopathy and tissue necrosis[17–19]. Two classes of SVMP-inhibitors have been historically investigated in the field of snakebite: metal chelators and peptidomimetic hydroxamate inhibitors[13]. These different molecules have distinct modes of action; chelators reduce the available pool of $Zn^{2+}$ required for SVMP bioactivity, while peptidomimetic hydroxamate inhibitors directly bind the $Zn^{2+}$ ion present in the catalytic core of the metalloproteinase[31]. Here, we compared the inhibitory capabilities of the peptidomimetic inhibitors marimastat and batimastat (both Phase 2-approved) and the chelators DMPS and dimercaprol (both licensed drugs) (Supplementary Fig. 1) against a variety of venoms representing highly medically important viper species from distinct geographical regions[15,32–34] and with variable toxin compositions (Fig. 1); namely the West African and south Asian saw-scaled vipers (*Echis ocellatus* and *Echis carinatus*), the Central American fer-de-lance or terciopelo (*Bothrops asper*), the African puff adder (*Bitis arietans*) and the south Asian Russell's viper (*Daboia russelii*).

We used an in vitro kinetic fluorogenic assay[28] to assess the SVMP bioactivity of each venom and its inhibition by varying concentrations of the four SVMP-inhibitors. All venoms exhibited considerable SVMP activity when compared to the PBS control (Fig. 2a), except for *D. russelii*, whose venom SVMP abundance was the lowest (6.9% of all venom proteins; Fig. 1) of

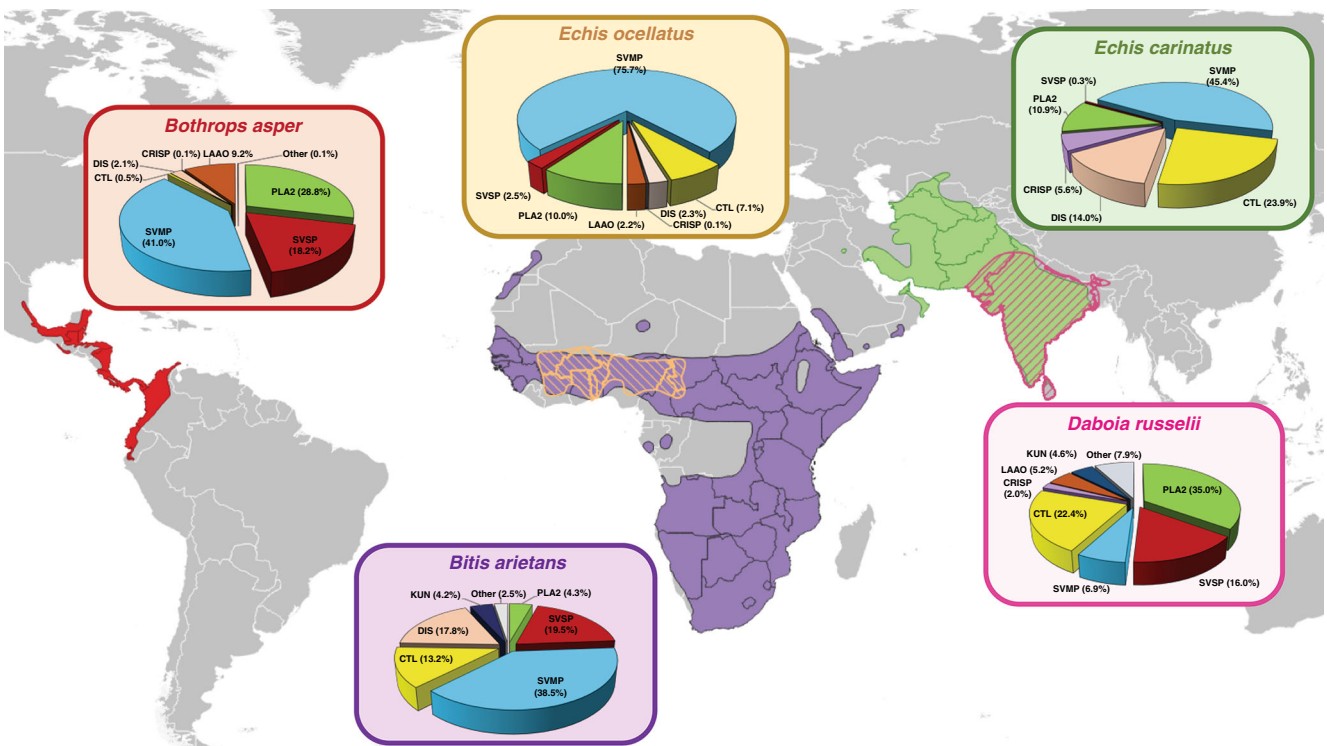

**Fig. 1 The geographical distributions and varying proteomic venom compositions of the medically important viper species used in this study.** The previously defined venom proteomes of *Echis ocellatus* (Nigeria)[63], *Echis carinatus* (India)[68], *Bothrops asper* (Costa Rica)[69], *Bitis arietans* (Nigeria)[70] and *Daboia russelii* (Sri Lanka)[64] are presented in pie charts. Toxin family key: SVMP snake venom metalloproteinase; SVSP snake venom serine protease; PLA2 phospholipase $A_2$; CTL C-type lectin; LAAO L-amino acid oxidase; DIS disintegrin; CRISP cysteine-rich secretory protein; KUN Kunitz-type serine protease inhibitor. Geographical species distributions were drawn using QGIS v3.10 software based on data downloaded from the World Health Organization Venomous Snake Distribution database and the IUCN Red List of Threatened Species database.

the tested species. Marimastat displayed complete in vitro neutralization of SVMP bioactivity of the four active venoms across a 1000-fold drug concentration range (150 nM to 150 µM) (Fig. 2b). Batimastat displayed equivalent efficacy to marimastat at the two lowest doses (1.5 µM and 150 nM, Fig. 2c), but could not be tested at the two higher concentrations due to the low water solubility of this drug and the interference of dimethyl sulfoxide (DMSO) (>1%) in our assay. Conversely, dimercaprol was generally effective down to 1.5 µM (IC$_{50}$s = 0.02–0.4 µM), and 15 µM of DMPS was required to fully inhibit SVMP activities (IC$_{50}$s = 0.29-4.19 µM) (Fig. 2b). We therefore concluded that both peptidomimetic inhibitors are equally effective (with IC$_{50}$s < 150 nM), and that both supersede the preclinically validated metal chelators[28] in neutralizing the in vitro SVMP activities of the African, Asian and American snake venoms tested here.

**Procoagulant venom activities are antagonized by peptidomimetic inhibitors and metal chelators.** Since SVMPs are key toxins associated with causing coagulopathy, we next investigated whether the same peptidomimetic and metal chelating inhibitors could also neutralize the procoagulant bioactivities of viper venoms. To do so, we used a validated kinetic absorbance-based assay monitoring plasma clotting[35] in the presence or absence of venoms and inhibitors (Fig. 3). Consistent with prior studies[36,37], all of the tested viper venoms displayed net procoagulant activity, with the exception of *B. arietans*, which had no effect on plasma clotting (Fig. 3a). While DMPS was effective at neutralizing the procoagulant activities of *E. ocellatus*, *E. carinatus* and *B. asper* venoms at the highest dose (150 µM) (Fig. 3b–d), it was ineffective against *D. russelii* (Fig. 3b). Dimercaprol outperformed DMPS in inhibiting procoagulant activity across all venoms

(IC$_{50}$ = < 0.15–3.7 µM vs 0.8–25.91 µM, respectively), whereas the two peptidomimetic inhibitors were equivalent to (*Echis* spp.) or outperformed (*B. asper* and *D. russelii*) both chelators across the concentration range tested (IC$_{50}$ = < 0.15–1.92 µM for marimastat and <0.15 µM for batimastat). Notably, the complete neutralization of SVMP activity in *D. russelii* venom by high doses of marimastat and batimastat revealed clear venom anticoagulant effects (Fig. 3b), which is likely due to the lack of neutralization of anticoagulant venom components, such as PLA$_2$s[37].

The results of the SVMP and coagulation assays demonstrated that the peptidomimetic inhibitors outperformed the metal chelators and, although marimastat and batimastat are similar drugs in terms of both mechanism of action and in vitro efficacy, marimastat has a number of potential clinical advantages over batimastat, including: (i) increased solubility, (ii) excellent oral bioavailability vs parenteral administration, and (iii) generally well tolerated vs some reports of acute bowel toxicity[38]. Therefore, we selected marimastat as our candidate SVMP-inhibitor for use in in vivo venom-neutralization experiments.

**Combinations of inhibitors inhibit distinct pro- and anticoagulant venom toxins.** The coagulation assay findings described above for *D. russelii* provided a strong rationale for exploring combinations of small molecule toxin inhibitors as snakebite therapeutics. While the SVMP-inhibitors potently inhibited the dominant procoagulant activities of this venom, inhibition revealed a secondary, uninhibited, anticoagulant activity (Fig. 3b). To better understand these effects, we applied a validated nanofractionation approach[35,37] to *D. russelii* venom and reassessed the inhibition of pro- and anticoagulant bioactivities of the resulting venom fractions (Supplementary Fig. 2A). Consistent

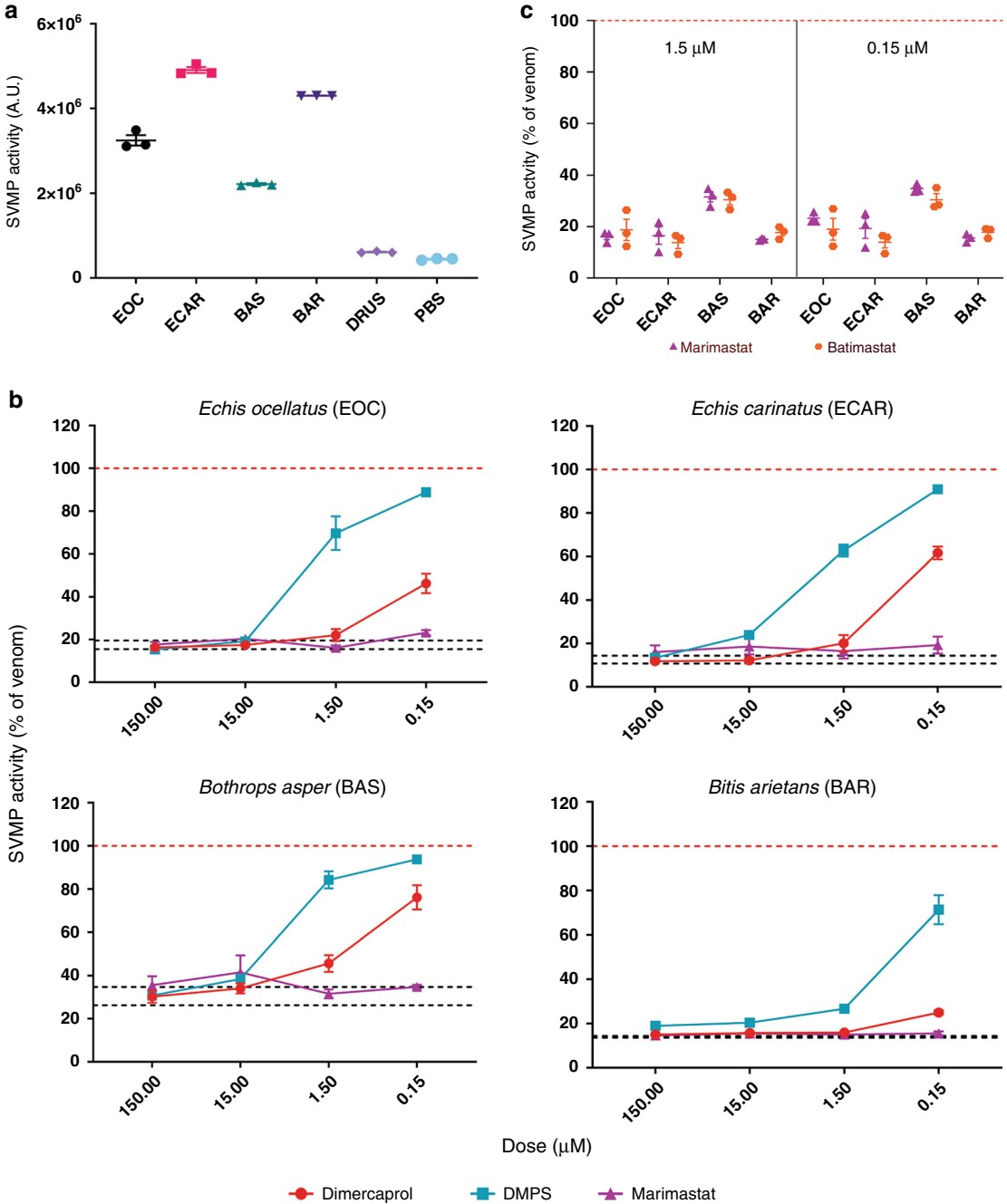

**Fig. 2 Small molecule toxin inhibitors inhibit the in vitro SVMP activities of several geographically distinct viper venoms. a** SVMP activities of the five viper venoms quantified by fluorogenic assay. The data presented represent mean measurements and SEMs of area under the curves of fluorescent arbitrary units taken from three independent experimental runs. EOC *E. ocellatus*; ECAR *E. carinatus*; BAS *Bothrops asper*; BAR *Bitis arietans*; DRUS *Daboia russelii*. **b** The effectiveness of metal chelators and peptidomimetic hydroxamate inhibitors at inhibiting the SVMP activity of the various viper venoms. Drug concentrations from 150 μM to 150 nM (highest to lowest dose tested) are presented. The data is expressed as percentage of the venom-only sample (100%, dashed red line). The negative control is presented as an interval (dashed black lines) and represents the values recorded in the PBS-only samples (expressed as percentage of venom activity), where the highest and the lowest values in each set of independent experiments are depicted. Inhibitors are color-coded (dimercaprol, red; DMPS, turquoise; marimastat, purple). **c** Comparison of SVMP inhibition by marimastat (purple) and batimastat (orange) at two concentrations (1.5 μM, left; 0.15 μM, right), expressed as the percentage of the venom-only sample (100%, dashed red line). All data represent triplicate independent repeats with SEMs, where each technical repeat represents the mean of $n \geq 2$ technical replicates. Source data provided in Supplementary Data S1.

with our findings using whole venom, the resulting nano-fractionated bioactivity profiles displayed procoagulant peaks that were effectively inhibited in a dose-dependent manner by marimastat, while fractions with anticoagulant activity were not

neutralized by this inhibitor at any of the tested concentrations (Supplementary Fig. 2A).

Prior research suggests that the anticoagulant activity of *D. russelii* venom is mediated by PLA$_2$ toxins[37]. Indeed, of the toxins

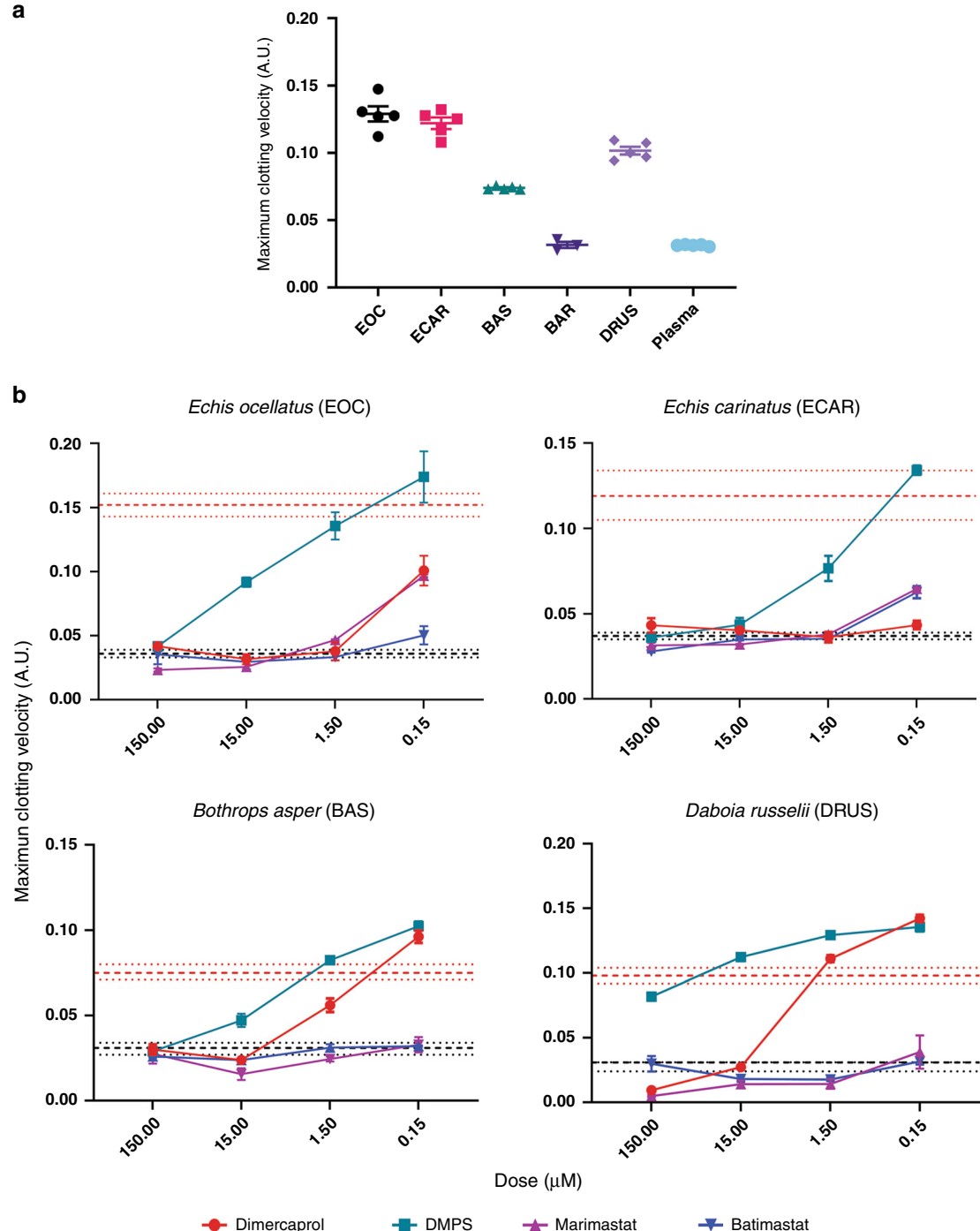

**Fig. 3 SVMP-inhibitors neutralize the in vitro procoagulant activities of several geographically distinct viper venoms. a** The coagulopathic activities of the viper venoms, showing that all, except *B. arietans* (BAR), exhibit procoagulant effects by increasing the clotting velocity in comparison with the normal plasma control. The data presented represents the maximum clotting velocity, calculated as the maximum of the first derivative of each clotting curve, from triplicate independent repeat experiments with SEMs, where each technical repeat represents the mean of $n \geq 2$ technical replicates. EOC *E. ocellatus*; ECAR *E. carinatus*; BAS *Bothrops asper*; BAR, *Bitis arietans*; DRUS, *Daboia russelii*. **b** Neutralization of procoagulant venom activity by four SVMP-inhibitors across four drug concentrations (150 μM to 150 nM). The data is expressed as the maximum clotting velocity at each dose. The negative (PBS) and positive (venom-only) controls are presented as intervals (dashed black and red lines, respectively), with the latter representing the mean maximum clotting velocity in these samples ± SEM. Inhibitors are color-coded (dimercaprol, red; DMPS, turquoise; marimastat, purple; batimastat, blue). The data represent triplicate independent repeats with SEMs, where each technical repeat represents the mean of $n \geq 2$ technical replicates. Note the different y-axis scales. Source data provided in Supplementary Data S1.

found in *D. russelii* venom, 35% are PLA$_2$s, while only 16% are SVSPs and 6.9% SVMPs[5] (Fig. 1). Consequently, we tested the well-established PLA$_2$-inhibitor varespladib (Supplementary Fig. 1) against the same venom fractions. As anticipated, we found that varespladib effectively inhibited the anticoagulant activity of *D. russelii* venom (Supplementary Fig. 2A) and, surprisingly, also exhibited some inhibitory effect against procoagulant venom toxins, but only at very high doses (20 μM), suggesting perhaps a non-specific effect (Supplementary Fig. 2A).

Since marimastat effectively neutralizes the SVMP-driven procoagulant activity of *D. russelii* venom, while varespladib inhibits the anticoagulant PLA$_2$ toxins, we next tested whether a combination of these two drugs could restore normal clotting caused by the whole venom. At the two highest doses tested (15 and 150 μM), the combination of these two inhibitors restored clotting profiles to levels similar to those observed in the control (Supplementary Fig. 2B), demonstrating that a rationally designed small molecule toxin inhibitor mix is capable of simultaneously inhibiting both procoagulant and anticoagulant venom toxins.

**Venom SVSP activities are abrogated by the serine protease-inhibitor nafamostat.** While inhibitors against SVMP and PLA$_2$ toxins have been actively researched, to our knowledge no serine protease-inhibitors have been investigated as drugs against snakebite. We selected nafamostat (Supplementary Fig. 1), a serine protease-inhibitor licensed as an anticoagulant medicine in Japan[39], as a candidate SVSP-inhibitor and tested its in vitro efficacy using a chromogenic assay. Among the tested venoms, all except *D. russelii* displayed detectable SVSP activity in our assay (Fig. 4a). These activities were broadly neutralized in a dose-dependent manner by nafamostat (Fig. 4b), with the highest doses (150 and 15 μM) completely inhibiting SVSP activity, irrespective of venom (IC$_{50s}$ = 0.12–1.07 μM). Although SVSP toxins can also perturb coagulation, we were unable to test the efficacy of nafamostat in the plasma assay described above due to nafamostat's inherent anticoagulant potency (Supplementary Fig. 3), which is mediated via interactions with cognate serine proteases found in the blood clotting cascade, such as thrombin and factors Xa and XIIa[40]. Because of these off-target interactions, generic SVSP-inhibitors must be carefully evaluated prior to any inclusion in a human snakebite therapy, especially since SVSPs are often less abundant in venom than SVMP or PLA$_2$ toxins[5] (Fig. 1). Nevertheless, the in vitro efficacy of nafamostat demonstrated here justified its evaluation in in vivo models of envenoming to select the most efficacious mixture of inhibitors.

**Preclinical efficacy of small molecule toxin inhibitors as solo and combination therapies.** We used an established in vivo model of envenoming[41,42] to test the efficacy of small molecule toxin inhibitors. This model consists of the preincubation of the test therapy with venom, followed by intravenous injection of the mixture into groups of five male CD-1 mice (18–20 g) via the tail vein, and is based on the gold standard method of preclinical efficacy recommended by the World Health Organization[43]. We first tested the ability of marimastat, varespladib and nafamostat as solo therapies to prevent venom-induced lethality in mice challenged with a 2.5× median lethal dose (LD$_{50}$) of *E. ocellatus* venom (45 μg)[25]. We selected this snake venom and venom dose as our initial model based upon its medical importance and results from our recent work exploring the preclinical venom-neutralizing efficacy of metal chelators[28]. All five of the experimental animals receiving only *E. ocellatus* venom succumbed to the lethal effects within 50 min. Both the PLA$_2$-inhibitor

varespladib and the SVMP-inhibitor marimastat (60 μg inhibitor/mouse) prolonged the survival of experimentally envenomed animals (Fig. 5a). However, marimastat conferred substantially greater protection than varespladib, as only one experimental animal succumbed towards the end of the experimental time frame of 6 h (death at 216 min), and the remaining four survived (Fig. 5a), while treatment with varespladib failed to prevent lethality over the full experimental time course, with two early deaths (5 and 9 min) and three later deaths (67, 210 and 341 vs <50 min for the venom-only control) observed (Fig. 5a). Conversely, the administration of the SVSP-inhibitor nafamostat (60 μg inhibitor/mouse) resulted in no evident efficacy, with negligible differences in survival times compared with the venom-only control (mean survival of 27.4 vs 17.8 min, respectively). Inhibitor-only controls revealed no obvious signs of acute toxicity of any of the drugs, as experimental animals survived without ill effects and exhibited normal behaviors throughout the 6 h treatment period.

We next tested the preclinical efficacy of two different inhibitor combinations against the lethal effects of *E. ocellatus* venom; (i) a dual mixture consisting of marimastat and varespladib (MV, 60 μg each) and (ii) a triple mixture containing marimastat, varespladib and nafamostat (MVN, 60 μg each). Both toxin inhibitor mixtures resulted in survival of all experimental animals until the end of the experiment (Fig. 5b), demonstrating that the combination of small molecules results in increases in efficacy, in line with our in vitro coagulation findings.

We next assessed markers of venom-induced coagulopathy in the envenomed animals via the quantification of thrombin-antithrombin (TAT) levels, a proxy for thrombin generation, in plasma collected following euthanasia. In line with previous reports[25,28], TAT levels correlated well with treatment efficacy. While animals in the venom-only group displayed very high TAT levels (mean of 1127.3 ng/ml), those receiving the MV and MVN inhibitor combinations exhibited substantially lower levels (162.0–203.9 and 238.1–255.5 ng/ml), and closer to those found in normal mice (i.e., no venom or treatment) controls (10.2–15.9 ng/ml) (Fig. 5c). TAT levels in the marimastat solo therapy group were also reduced (231.2–267.9 ng/ml) and comparable with the two combination therapies, but those detected in the less efficacious varespladib- and nafamostat-only treatment groups displayed substantially higher TAT levels (472.7–649.9 ng/ml and 543.9–859.1 ng/ml, respectively), although these remain lower than those of the venom-only controls. In combination, these findings suggest that marimastat is likely responsible for much of the observed efficacy against the lethal effects of *E. ocellatus* venom, but that small molecule combinations with additional toxin inhibitors provide superior preclinical efficacy than treatment with marimastat alone.

**Inhibitor mixes protect against lethality caused by a diverse range of viper venoms.** We next investigated whether the two inhibitor combination therapies were equally effective against the other viper venoms tested in vitro, as these venoms exhibit highly variable toxin compositions in comparison with the SVMP-rich toxin profile of *E. ocellatus* (Fig. 1). We adopted the same approach as described above, and intravenously challenged groups of experimental animals with 2.5 × LD$_{50}$ doses of *E. carinatus* (47.5 μg)[41], *B. asper* (47 μg)[44], *B. arietans* (54 μg)[41] and *D. russelii* (20 μg)[45] venoms in the presence and absence of the MV and MVN therapeutic mixtures.

These results of these studies demonstrate the therapeutic potential of small molecule inhibitors, as despite extensive venom differences, we found that the dual mixture of marimastat and varespladib protected mice from the lethal effects of all four

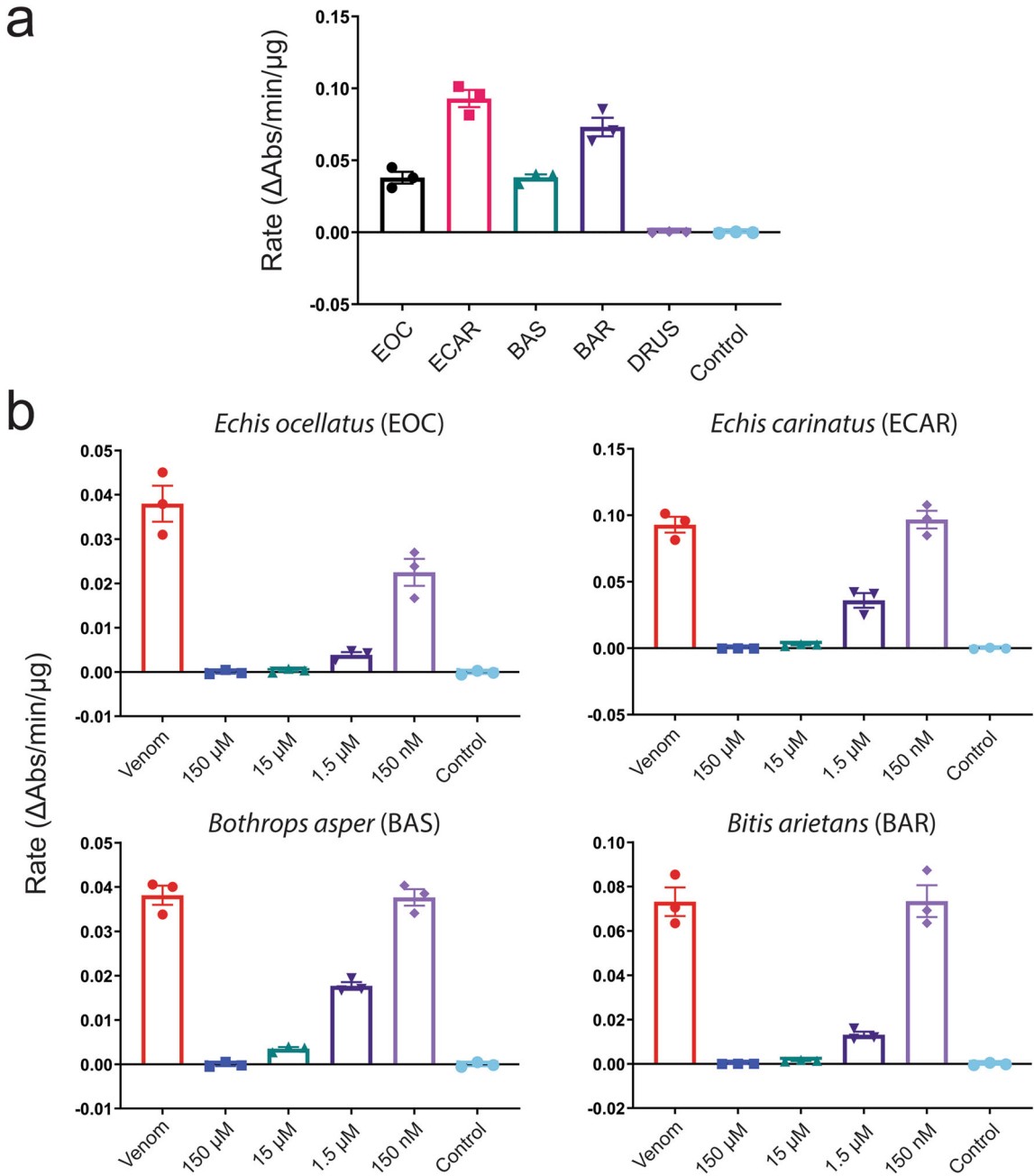

**Fig. 4 Nafamostat inhibits the in vitro serine protease activities of several geographically distinct viper venoms. a** The serine protease (SVSP) activity of five viper venoms expressed as the rate (ΔAbs/time/μg venom) of substrate consumption determined by kinetic chromogenic assay. The data represents triplicate independent repeats with SEMs, where each technical repeat represents the mean of $n \geq 2$ technical replicates. EOC *E. ocellatus*; ECAR *E. carinatus*; BAS *Bothrops asper*; BAR *Bitis arietans*; DRUS *Daboia russelii*. **b** Neutralization of SVSP venom activity by the serine protease-inhibitor nafamostat. The data is expressed as rates (ΔAbs/time/μg venom) and represents triplicate independent repeats with SEMs, where each technical repeat represents the mean of $n \geq 2$ technical replicates. Venom only activity (venom) is displayed alongside venom incubated with decreasing molarities of nafamostat (150 μM to 150 nM). Note the different y-axis scales. Source data provided in Supplementary Data S1.

venoms for the duration of the experiment (Fig. 5d–g). The triple mixture, additionally containing nafamostat, proved equally effective across the venoms, with the exception of one early death (31 min) in the group dosed with *D. russelii* venom (Fig. 5d–g). TAT levels increased in all venom-only groups (Fig. 5h), although these increases were negligible in those receiving *B. arietans* venom—a finding in line with our in vitro data suggesting that this venom has little coagulopathic activity (Fig. 3a). TAT levels were consistently reduced in the experimental animals treated with the two inhibitor mixtures, resulting

in 53.6–95.5% reductions compared with the various venom-only groups (Fig. 5h).

Comparable preclinical efficacy between the MV and MVN inhibitor combinations against a variety of medically important viper venoms suggests that the SVSP-inhibitor nafamostat does not contribute substantially to venom neutralization. Because (i) SVSP-inhibitors such as nafamostat can induce off-target effects by interacting with serine proteases found in the coagulation cascade, (ii) the inclusion of every additional molecule in an inhibitory therapeutic combination substantially increases

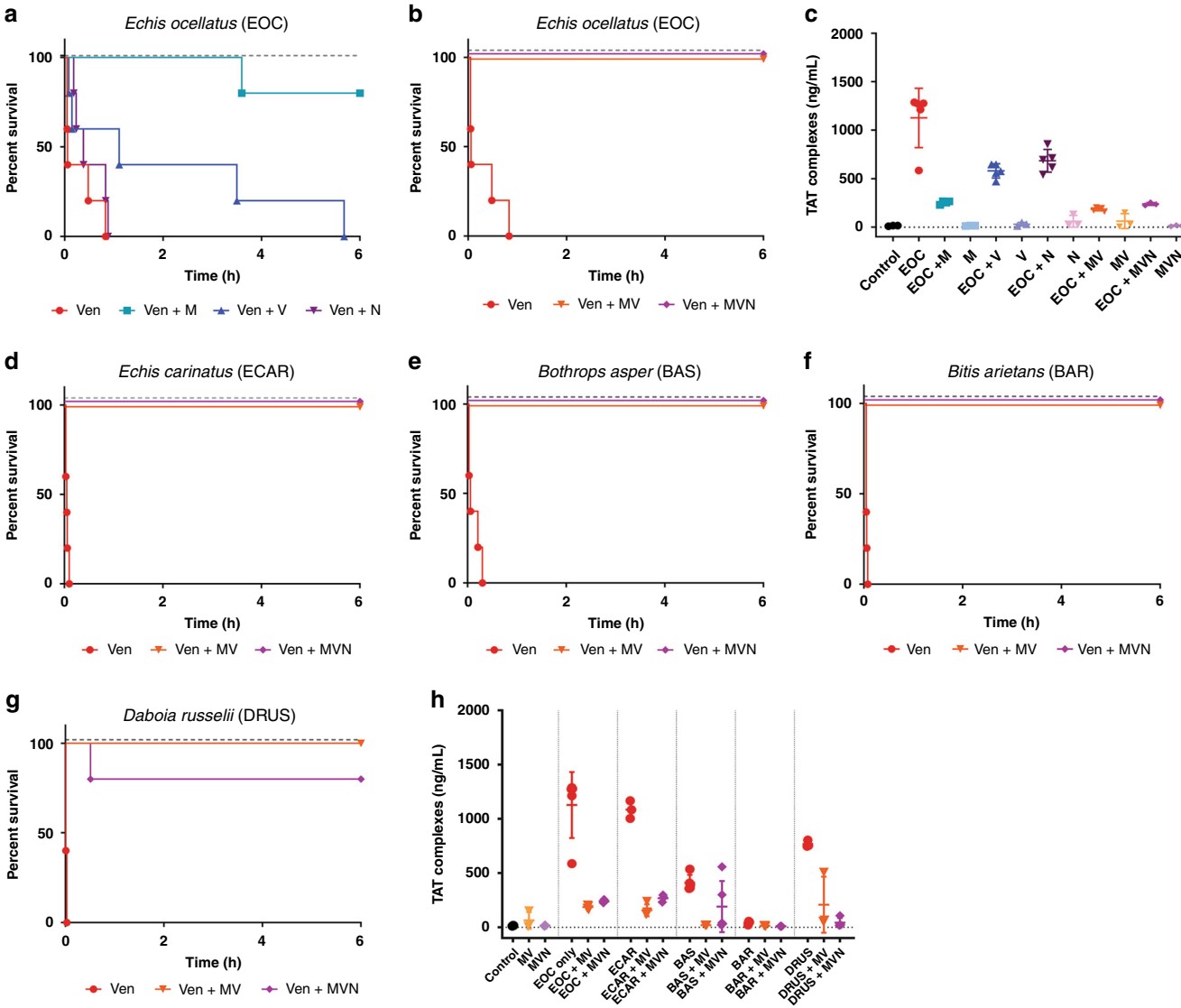

**Fig. 5 Combinations of small molecule toxin inhibitors broadly protect against venom lethality in an in vivo 'preincubation' model of snake envenoming.** Kaplan–Meier survival graphs for experimental animals ($n = 5$) receiving venom (Ven) preincubated (30 min at 37 °C) with different small molecule inhibitors or inhibitor mixes via the intravenous route and monitored for 6 h. Drug-only controls are presented as black dashed lines at the top of each graph (none of the drugs exhibited any observable toxicity at the given doses). **a** Survival of mice receiving 45 μg of *E. ocellatus* venom ($2.5 \times LD_{50}$ dose) with and without 60 μg of marimastat or varespladib or nafamostat. **b** Survival of mice receiving 45 μg of *E. ocellatus* venom ($2.5 \times LD_{50}$ dose) with and without a dual combination mixture of marimastat and varespladib (MV, 60 μg each) or a triple combination mixture of marimastat, varespladib and nafamostat (MVN, 60 μg each). **c** Quantified thrombin-antithrombin (TAT) levels in the envenomed animals from (**a**) and (**b**). Where the time of death was the same within experimental groups (e.g., early deaths or complete survival) TAT levels were quantified for $n = 3$, and where times of death varied, $n = 5$. The data displayed represents means of the duplicate technical repeats plus SDs. **d–g** Kaplan–Meier survival graphs for experimental animals ($n = 5$) receiving inhibitor mixes (MV or MVN) preincubated with $2.5 \times LD_{50}$ dose of *E. carinatus* (47.5 μg, **d**), *B. asper* (47 μg, **e**), *B. arietans* (54 μg, **f**) or *D. russelii* (20 μg, **g**) venom. **h** Quantified TAT levels in the envenomed animals from (**d**) to (**g**), with data presented as described for (**c**). Source data provided in Supplementary Data S1.

regulatory hurdles for future translation, and (iii) the inhibition of SVMP and PLA₂ toxins appears sufficient to protect against lethality caused by a diverse array of viper venoms, we decided to proceed with the marimastat and varespladib combination as our lead candidate for testing in more therapeutically challenging preclinical models of envenoming.

**Administration of the marimastat and varespladib (MV) dual therapy after venom challenge broadly protects against venom lethality.** To better mimic a real-life envenoming scenario, we next tested the marimastat and varespladib inhibitor mixture in a

preclinical 'challenge then treat' model of envenoming, where the venom is first administered intraperitoneally and then the test therapy is administered intraperitoneally separately after the venom challenge[28]. To this end, we injected venom from each of the five viper species in doses equivalent to at least 5× the intravenous (i.v.) $LD_{50}$ dose followed, 15 min later, by a single dose of the inhibitor mixture (120 μg of both marimastat and varespladib). Experimental animals were then monitored for 24 h. For *E. ocellatus*, *E. carinatus* and *B. arietans* venoms we challenged mice with 5× i.v. $LD_{50}$ doses (90, 95, and 108 μg, respectively), while higher venom doses were required for *B. asper* (303 μg, equivalent to ~16× i.v. $LD_{50}$ or 3× i.p. $LD_{50}$[46]) and *D. russelii*

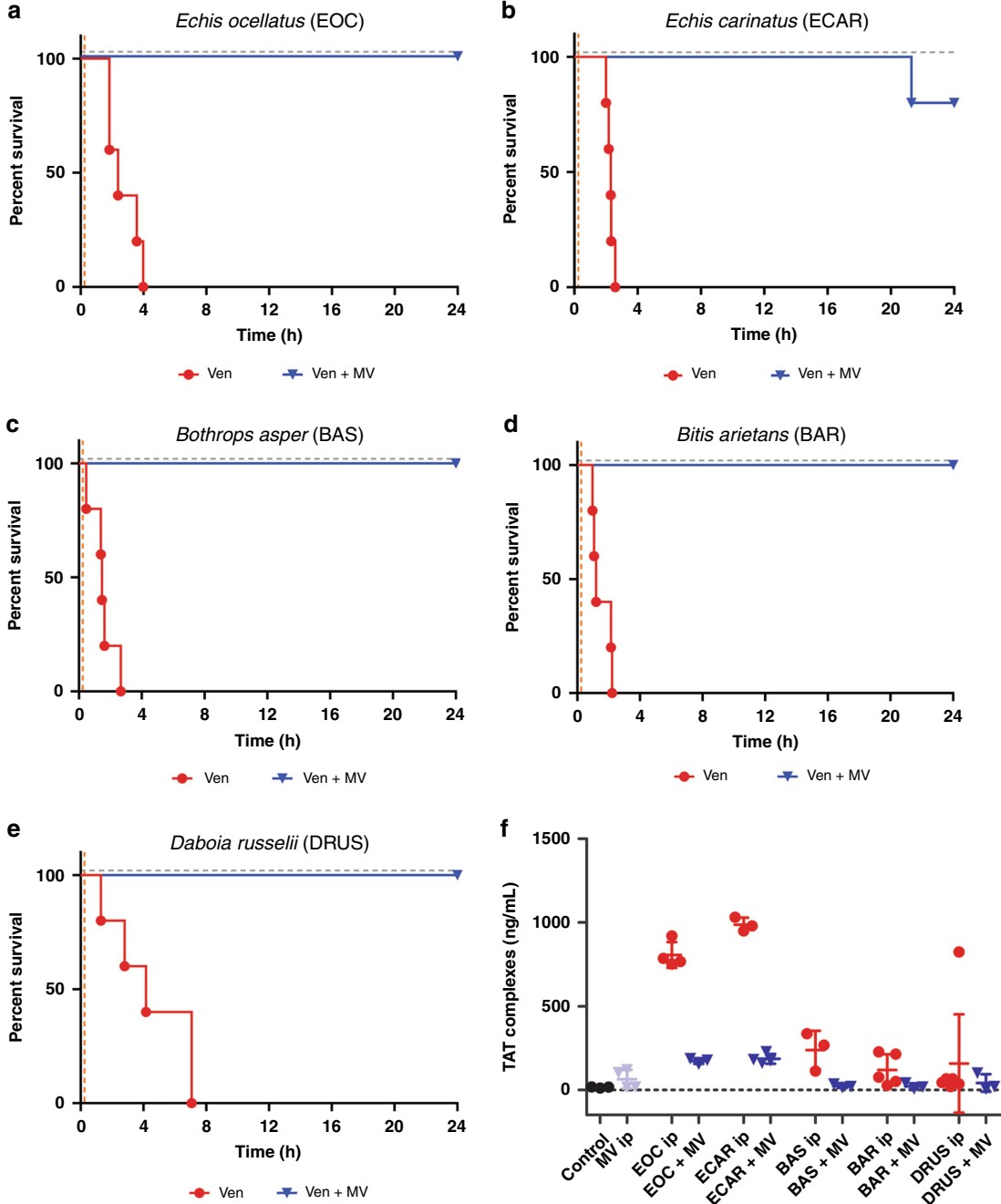

**Fig. 6 The inhibitor combination of marimastat and varespladib (MV) provides broad preclinical efficacy against venom lethality in an in vivo 'challenge then treat' model of envenoming.** Kaplan–Meier survival graphs for experimental animals ($n = 5$) receiving venom (Ven), followed by delayed drug treatment (15 min later) with a dual combination of marimastat and varespladib. Both venom and treatment were delivered via the intraperitoneal route, and the end of the experiment was at 24 h. Survival of mice receiving: **a** *E. ocellatus* (90 μg, 5× i.v. $LD_{50}$), **b** *E. carinatus* (95 μg, 5× i.v. $LD_{50}$), **c** *B. asper* (303 μg, 3× i.p. $LD_{50}$), **d** *B. arietans* (108 μg, 5× i.v. $LD_{50}$) and **e** *D. russelii* (105 μg, 13× i.v. $LD_{50}$) venom, with and without the inhibitor mix (120 μg of each drug) 15 min later. The drug-only control is presented as a black dashed line at the top of each graph (no toxicity was observed at the given dose). **f** Quantified thrombin-antithrombin (TAT) levels in the envenomed animals from (**a**) to (**e**). Where the time of death was the same within experimental groups (e.g., early deaths or complete survival) TAT levels were quantified for $n = 3$, and where times of death varied, $n = 5$. The data displayed represents means of the duplicate technical repeats plus SDs. Source data provided in Supplementary Data S1.

($13×$ i.v. $LD_{50}$, 105 μg) to ensure mortality occurred within 7 h, thus leaving a 17 h window for measuring prolonged survival in the treatment groups.

All of the venom-only groups succumbed to the lethal effects of envenoming within 4 h, with the exception of two mice receiving *D. russelii* venom (deaths at ~7 h), while experimental animals dosed with only the inhibitor combination (i.e., treatment

control) survived the duration of the experiment (24 h) with no apparent adverse effects (Fig. 6a–e). Across all of the diverse venoms tested, the delayed administration of a single dose of the marimastat and varespladib combination resulted in prolonged survival for at least 17 h after the venom-only controls suffered venom-induced lethality (Fig. 6a–e). All animals receiving the delayed treatment survived for the full duration of the experiment

(24 h) irrespective of the venom used as challenge, with the sole exception of one mouse receiving *E. carinatus* venom, for which survival was still prolonged by at least 18 h when compared with the venom-only control (death registered at 21.3 h) (Fig. 6b).

Quantified TAT levels from the envenomed animals correlated with survival, with those receiving the inhibitor mixture exhibiting 78.5–90.5% reductions compared to the elevated levels of the various venom-only controls (Fig. 6f). Contrastingly, quantification of soluble thrombomodulin, a marker of endothelial cell damage, was only elevated in *B. asper* 'envenomed' mice. This observation was noted for both the 'intravenous preincubation' and 'intraperitoneal challenge then treat' models of envenoming (Supplementary Fig. 4A, B), and these elevated levels were reduced to control levels in experimental animals treated with the inhibitor combination in both experimental approaches (Supplementary Fig. 4C). These findings suggest that, in addition to protecting against the lethal effects of the various viper venoms, the marimastat and varespladib therapeutic combination is capable of preventing coagulopathy, and in the case of *B. asper*, inhibiting toxins acting to disrupt certain components of the endothelium.

## Discussion

Snakebite is the world's most lethal NTD, resulting in ~138,000 deaths annually and primarily affecting the world's resource-poor populations of the tropics and subtropics[1]. Although conventional polyclonal immunoglobulin-based antivenoms save thousands of lives each year, their lack of specificity, poor cross-species efficacy, reliance on delivery in clinical settings and low affordability severely hamper their accessibility and utility for treating tropical snakebite victims[1,42]. Consequently, new strategies capable of circumventing variation in snake venom composition to deliver broad neutralization across snake species, while simultaneously improving the safety, affordability and storage logistics of treatment, are urgently needed[3,47]. Approaches showing signs of promise include the rational design of immunogens to improve the neutralizing breadth of conventional products[48], the selection of human or humanized toxin-specific monoclonal or oligoclonal antibodies[49,50], and the use of small molecule inhibitors specific to certain toxin families[12], such as the PLA$_2$-inhibitor varespladib[22,26,27] and the metal chelator DMPS[28]. Small molecule toxin inhibitors offer a number of desirable characteristics over existing snakebite therapies, including desirable specificity, potent dose-efficacy, higher tolerability, greater stability and superior affordability[12]. These characteristics, combined with their oral formulation, provide an opportunity to explore their utility as prehospital treatments for snakebite, thereby circumventing one of the major challenges faced by impoverished snakebite victims, who have great difficulty in rapidly accessing the secondary and tertiary healthcare facilities where current treatments are held. Here, we show that a small molecule mixture consisting of the inhibitors marimastat and varespladib, which are directed against the hemotoxicity-inducing SVMP and PLA$_2$ toxin families, provides preclinical protection against lethality caused by a geographically diverse array of medically important viper venoms that differ considerably in their toxin compositions.

Due to their importance in many snake venoms, we first rationally selected an inhibitory molecule capable of abrogating the activity of SVMP toxins. In vitro SVMP and coagulation assays convincingly demonstrated that the Phase 2-approved peptidomimetic hydroxamate inhibitors batimastat and marimastat provided superior venom neutralization over the metal chelators DMPS and dimercaprol (Figs. 2 and 3). Despite previous reports of batimastat exhibiting increased efficacy over

marimastat in preventing venom-induced local hemorrhage[23], we found both drugs to be equipotent in vitro. We selected marimastat as our candidate for in vivo efficacy experiments due to a number of desirable characteristics that make it amenable for a future field intervention for snakebite, specifically its oral vs intraperitoneal route of administration, and its increased solubility and tolerability compared to batimastat[38]. Indeed, these characteristics seemingly contributed to the demise of batimastat during development, although both drugs were ultimately discontinued following lack of efficacy in Phase 3 clinical trials[38], despite showing early promise as cancer therapeutics[51]. Marimastat displays particularly good oral bioavailability. It can be detected in the blood of patients within 15–60 min after ingestion, reaches peak plasma concentrations 1.5–3 h post-administration with a half-life of 8–10 h[52], and can be detected in the circulation when given at doses >200 mg for up to 2 days[52]. Furthermore, marimastat is well tolerated, with no notable side effects observed with single doses of up to 800 mg or bidaily doses of 200 mg for 6.5 days[52], or when 75 mg doses were administered daily for 28 days in patients with advanced pancreatic cancer[53]. In our study, we used a low dose of 3 or 6 μg/g for the intravenous and intraperitoneal murine models of envenoming respectively, which translates to 0.24 and 0.48 mg/kg when applying a facile mouse to human dose conversion[54]. Even when considering the differences in route of administration (intravenous/intraperitoneal vs oral), our extrapolated dose (33.6 mg per 70 kg adult) is very low compared to that well tolerated in Phase 1 trials (800 mg) and this, combined with the relatively high oral bioavailability of marimastat (70%), offers substantial scope for the development of this drug as a prehospital therapeutic for use soon after a snakebite. However, murine bridging studies incorporating pharmacokinetic (PK) profiling coupled with pharmacodynamic (PD) assessments of venom neutralization are required in the future to enable accurate simulations of predicted human doses.

The second drug in our mixture, varespladib, is a secretory PLA$_2$-inhibitor previously investigated for use in the treatment of various acute coronary syndromes[55]. Both varespladib and varespladib methyl (its oral prodrug, which is rapidly converted in vivo to varespladib) have been used clinically in Phase 1 and 2 trials[55–57], although a lack of efficacy at Phase 3 ultimately resulted in discontinuation[58]. More recently, varespladib has been explored for repurposing as a potential therapeutic for the treatment of snakebite. Both varespladib and its oral prodrug have been shown to exhibit promising neutralizing capabilities against a variety of different snake venoms[22,29], but have proven to be particularly effective at mitigating the life-threatening effects of neurotoxicity caused by certain elapid venoms in animal models of envenoming[26,27,59]. Similar to marimastat, varespladib shows good oral bioavailability, and has a half-life equating to 5 h when delivered by iv infusion[55]. Varespladib has also been demonstrated to be well tolerated at Phase 1 and 2[55], although a double-blind randomized Phase 3 clinical trial showed that acute coronary syndrome patients receiving 500 mg of oral varesapldib daily had a greater risk of myocardial infarction than those receiving placebo[58], despite the same daily dose used in Phase 2B (>300 patients for >6 months) resulting in no greater risk of major adverse cardiovascular events[56]. Given that the dose of varespladib used in these clinical studies is ~15-fold higher than the facilely-extrapolated human equivalent dose used intraperitoneally in our animal model (0.48 mg/kg, 33.6 mg per 70 kg adult)[54], there appears to be considerable space to safely optimize the dose and dosing frequency of varespladib to establish an appropriate therapeutic regimen for use for treating snakebite, though murine bridging study-based simulations of appropriate human doses are needed.

Our in vivo venom neutralization studies demonstrate that a combination of these SVMP- and PLA$_2$-inhibiting drugs is capable of counteracting the lethal hemorrhagic, coagulopathic and/or hemostasis-disrupting effects of a variety of venoms sourced from the most medically important vipers of Central America, sub-Saharan African and South Asia[15,32–34] (Figs. 5 and 6). The addition of the serine protease-inhibitor nafamostat to the therapeutic mixture resulted in no additional protection to the marimastat and varespladib dual combination (Fig. 5) despite SVSP toxins also being common pathogenic constituents of many viper venoms[5], and this drug exhibiting potent inhibition of SVSP toxins in vitro (Fig. 4). These findings, alongside evidence that nafamostat provides no protection against the lethal effects of *E. ocellatus* venom when used as a solo therapy (Fig. 5a), suggest that nafamostat does not appear to substantially contribute to the preclinical efficacy observed (Fig. 5). Despite being a licensed anticoagulant drug in Japan since the early 1990s[39], nafamostat has potential detrimental off-target effects for use in snake envenoming via interaction with cognate coagulation cascade serine proteases[40], has a short half-life (~8 min)[60], and requires intravenous administration, thereby limiting its utility and applicability as a potential prehospital snakebite therapeutic. For those various reasons, our lead candidate therapeutic mixture remained restricted to the marimastat and varespladib combination.

The administration of the marimastat and varespladib combination 15 min after 'envenoming' resulted in the survival of experimental animals for at least 17 h after mortality was observed in the venom-only control groups (Fig. 6). In our previous work, we demonstrated that the licensed metal chelator DMPS, which shows much promise as an early intervention therapeutic against snakes with SVMP-rich venoms (e.g., the West African saw-scaled viper, *E. ocellatus*)[28], prevented lethality for ~8 h in the same preclinical model, but required a later dose of antivenom (1 hr after venom delivery) to extend protection to a comparable duration to that observed here with the marimastat and varespladib combination (Supplementary Fig. 5). While DMPS remains a promising future therapeutic for snakebite, not least because of its oral formulation, licensed drug status and decades of therapeutic use for other indications[61,62], it seems unlikely to be highly efficacious as a solo therapy against a wide variety of different snake species due to only targeting SVMP toxins[28]. Contrastingly, the combination of marimastat and varespladib reported here provided consistent and prolonged preclinical protection against lethality caused by a wide diversity of medically important vipers despite, for example, the south Asian Russell's viper (*D. russelii*) having substantially different abundances of distinct venom toxins to that of *E. ocellatus*[4,63–65] (Fig. 1). Given that all existing antivenoms are geographically-restricted in terms of their snake species efficacy (e.g., restricted to specific continents or countries within), and require considerably higher doses to be preclinically effective (e.g., 166.66 µg of monospecific antivenom antibodies[25] vs 1.33 µg of each inhibitor per 1 µg of venom challenge for *E. ocellatus*, for example), these findings suggest that this therapeutic combination of small molecule toxin inhibitors may represent a highly specific yet generic future treatment for viperid snakebite.

Notwithstanding the apparent therapeutic promise of this small molecule toxin inhibitor combination, a considerable amount of future research is required to facilitate its translation. Despite the combination of animal models used here providing confidence of broad anti-envenoming efficacy, these models remain limited in terms of accurately recapitulating cases of human envenoming (e.g., in terms of venom dose, route of venom delivery, treatment duration, etc). Thus, additional preclinical studies are needed to further explore the neutralizing efficacy of this drug combination, including the use of oral dosing regimens, and repeat dosing experiments combined with pharmacokinetic analyses, to effectively model the oral dose required to maintain effective concentrations of the drugs sufficient to provide prolonged protection from envenoming. This may be particularly challenging for cases where envenoming may result in prolonged treatment times, for example, as the result of recurrence of coagulopathy or acute kidney injury, and thus additional model development addressing this point is needed. While no overt adverse reactions were observed in the experimental animals used in this study, and both marimastat and varespladib have previously been demonstrated to be well tolerated clinically[52,53,57,66], potential drug-drug interactions at PK/PD-informed human doses also need to be robustly assessed. The in vitro and in vivo venom neutralization data presented here should also be extended to additional medically important snake species, and the efficacy of this combination therapy against the local, morbidity-inducing, effects of snake venoms should be explored. Finally, the successful delivery and uptake of any prehospital snakebite treatment comes with a number of long-term implementation challenges that require careful consideration, including ensuring (i) acceptable safety profiles across the target population (e.g., both children and adults) and (ii) that health seeking behavior after initial treatment is strongly promoted so that patients are carefully monitored in case additional (i.e., doses) or complementary (i.e., antivenom) treatment is required.

Despite these remaining challenges, we demonstrated here that a combination of two Phase 2-approved drugs, marimastat and varespladib, provides broad protection against venom-induced lethality in both a conventional 'preincubation' model of envenoming, and a far more challenging preclinical model consisting of delayed drug delivery post-envenoming. While these findings hold much promise, we propose that the future translation of this inhibitor combination should occur in parallel with other small molecule toxin inhibitor lead candidates, such as DMPS and varespladib[26–28,59], to increase the breadth of new molecules being added to the snakebite treatment toolbox and, most importantly, to help offset the risk of potential drug failures during clinical trials. Ultimately, our data provides the first empirical evidence that combinations of small molecule toxin inhibitors can provide cross-species neutralization of medically important snake venoms, and thus advocates for the future translation of such combinations as generic, prehospital treatments to reduce the life-threatening and life-changing consequences of the world's most lethal neglected tropical disease - snakebite.

## Methods

**Venoms**. Venoms were sourced from either wild-caught specimens maintained in, or historical venom samples stored in, the Herpetarium of the Liverpool School of Tropical Medicine. This facility and its protocols for the expert husbandry of snakes are approved and inspected by the UK Home Office and the LSTM and University of Liverpool Animal Welfare and Ethical Review Boards. The venom pools were from vipers with diverse geographical localities, namely: *E. ocellatus* (Nigeria), *E. carinatus sochureki* (India, referred to throughout as *E. carinatus*), *B. arietans* (Nigeria), *B. asper* (Atlantic coast of Costa Rica) and *D. russelii* (Sri Lanka). Note that the Indian *E. carinatus* venom was collected from a single specimen that was inadvertently imported to the UK via a boat shipment of stone, and then rehoused at LSTM on the request of the UK Royal Society for the Prevention of Cruelty to Animals (RSPCA). Crude venoms were lyophilized and stored at 4 °C to ensure long-term stability. Prior to use, venoms were resuspended to 10 mg/ml in PBS (pH 7.4) and then further diluted to 1 mg/ml stock solutions (with PBS) for the described experiments.

**Inhibitors**. Dimercaprol (2,3-dimercapto-1-propanol ≥98% iodometric, Cat no: 64046-10 ml), marimastat (≥98% HPLC M2699-5MG), batimastat (SML0041-5MG) and varespladib (≥98% HPLC SML1100-5MG) were purchased from Sigma-Aldrich. Nafamostat mesylate (ab141432 10 mg) was purchased from Abcam and DMPS (2,3-dimercapto-1-propanesulfonic acid sodium salt monohydrate, 95%,

Cat no: H56578) from Alfa Aesar. Working stocks (tenfold dilutions from 2 mM to 2 μM) were made using deionized water, with the exception of varespladib and batimastat, for which we used DMSO due to water insolubility.

**Enzymatic assays**. The SVMP assay measuring metalloproteinase activity and the plasma assay measuring coagulation in the presence or absence of venoms and inhibitors were performed as previously described[28]. The SVMP assay kinetically measured cleavage of a quenched fluorogenic substrate (ES010, R&D Biosystems) by venom in the presence or absence of inhibitors.

Briefly, 10 μl of the substrate (supplied as a 6.2 mM stock) was used per 5 ml reaction buffer (150 mM NaCl, 50 mM Tris-Cl pH 7.5). Reactions consisted of 10 μl of venom ± inhibitors in PBS and 90 μl of substrate. Venoms were used at 1 μg/reaction and final inhibitor concentrations ranged from 150 μM to 150 nM. Samples were preincubated for 30 min at 37 °C and pipetted in triplicate into 384-well plates (Greiner). Thereafter, data was collected on an Omega FLUOstar (BMG Labtech) instrument at an excitation wavelength of 320 nm and emission wavelength of 405 nm at 25 °C for 1 h. The areas under the curve (AUCs) in the 0–40 min interval were calculated for each sample using MARS data analysis software v3.31 (BMG Labtech); this time point was chosen as the time where all fluorescence curves had reached a plateau (maximum fluorescence). For comparing venom-only samples, the means of at least three independent experimental runs for each condition, expressed as AUCs ($n \geq 3$), were plotted at each inhibitor concentration with standard error of the mean (SEM). To determine inhibitor efficacy, the AUCs for each of the samples that consisted of venom + inhibitors were transformed and expressed as percentages of the venom-only sample (where the venom was 100%). The negative control (PBS-only) was also expressed relative to the venom, and the variation in background was presented as an interval delineated by the lowest and highest values in the PBS-only samples across concentrations and inhibitors for a specific venom. Due to the sensitivity of the assay to DMSO concentrations above 1% in the final reaction, the 2 mM stock of batimastat was further diluted in reaction buffer to obtain 200, 20, and 2 μM stocks, alongside appropriate DMSO only controls. However, only the latter two concentrations were sufficiently depleted of DMSO for viable use in the assay.

We used a previously developed plasma clotting assay[35] to measure venom-induced coagulation. Briefly, 100 ng of each venom was incubated at 37 °C for 30 min in the presence or absence of inhibitors (150 μM to 150 nM final concentrations). The final reaction consisted of 10 μl venom ± inhibitors in PBS, 20 μl 20 mM CaCl₂, and 20 μl citrated bovine plasma (VWR). The samples were pipetted in triplicate into 384-well plates, and absorbance monitored at 595 nm for 2 h at 25 °C on an Omega FLUOstar instrument. We calculated the maximum clotting velocity of each of the curves as per clot waveform analysis[67], by calculating the maximum of the first derivative. The means of at least three independent experimental runs for each condition were plotted at each inhibitor concentration with SEMs.

SVSP activity was measured using a chromogenic substrate (S-2288, Cambridge Bioscience). Reactions consisted of 15 μl of venom ± inhibitors in PBS, 15 μl of assay buffer (100 mM Tris-Cl pH 8.5, 100 mM NaCl) and 15 μl of 6 mM substrate. A positive venom-only control, a negative control containing 15 μl of PBS and a drug-only control (where applicable) were also included. The final substrate concentration was 2 mM and venoms were used at 1 μg/reaction. When testing inhibitors, these were preincubated with the venoms for 30 min at 37 °C with concentrations ranging from 150 μM to 150 nM (tenfold dilutions). The samples were pipetted in triplicate into 384-well plates (Greiner) and absorbance at 405 nm was monitored kinetically for ~30 min on an Omega FLUOstar instrument (BMG Labtech). Given the linearity of the resulting slopes, negative control readings (PBS) were subtracted from each reading and the rate of substrate consumption calculated by measuring the slope between 0 and 5 min. The mean rates (expressed as ΔAbs/time/μg venom) of at least three independent experimental runs for each condition ($n \geq 3$) were plotted at each inhibitor concentration with SEMs using Prism v8 software (GraphPad).

**Nanofractionation experiments**. Fifty microlitres of venom solution (D. russelii; 1 mg/ml) was injected into a Shimadzu LC-2010 system with nanofractionation in parallel. Separation was performed on a Waters XBridge reversed-phase C18 column (250 × 4.6 mm² column with 3.5 μm pore-size) at 30 °C. The total flow rate of the mobile phase solution was 0.5 ml/min with eluent A (98% H₂O, 2% acetonitrile [ACN], 0.1% formic acid [FA]) and eluent B (98% ACN, 2% H₂O, 0.1% FA). Liquid chromatography gradients consisted of a linear increase of mobile phase B from 0 to 50% in 20 min, followed by a linear increase to 90% B in 4 min, then isocratic at 90% B for 5 min, after which the percentage of mobile phase B was decreased from 90 to 0% in 1 min, followed by 10 min at 0% B to re-equilibrate. The effluent was split in a 1:9 ratio of which the 10% fraction was sent to a Shimadzu SPD-M20A prominence diode array detector and the 90% fraction was directed to a nanofractionation collector that dispensed the fractions into transparent 384-well plates (F-bottom, rounded square well, polystyrene, without lid, clear, non-sterile; Greiner Bio One, Alphen aan den Rijn, The Netherlands) at a resolution of 6 s/well. The collector used was either a commercially available FractioMate™ nanofractionator (SPARK-Holland & VU, Netherlands, Emmen & Amsterdam) controlled by FractioMator software v1.0 (Spark-Holland, The Netherlands, Emmen) or a modified Gilson 235P autosampler controlled by in-

house written software Ariadne v1.8 (VU Amsterdam). The well plates with venom fractions were then dried overnight in a Christ Rotational Vacuum Concentrator (RVC 2 − 33 CD plus, Zalm en Kipp, Breukelen, The Netherlands) equipped with a cooling trap, and maintained at −80 °C during operation. The freeze-dried plates were then stored at −80 °C prior to bioassaying.

Neutralization of coagulopathic venom toxins by marimastat and varespladib was assessed by assaying the D. russelii venom nanofractionated plates in the plasma coagulation assay, as recently described[35]. To each well of the nanofractionated well plate, 10 μl of inhibitor solution (e.g., marimastat or varespladib in PBS, or PBS-only control) was pipetted by a VWR Multichannel Electronic Pipet, followed by brief low-speed collection of samples via centrifugation at 805 × g. The final assay concentrations of the inhibitors tested in the assay were 20, 4, 0.8, 0.16, 0.032, and 0.0064 μM. The plates were then incubated for 30 min at room temperature, and during this time bovine plasma (Sterile Filtered, Biowest, Nuaille, France) was defrosted in a water bath and centrifuged for 4 min at 805 × g prior to use. Following incubation, 20 μl CaCl₂ solution (20 mM), followed by 20 μl plasma (with instrument rinsing in between with Milli-Q water), were pipetted into each well on the plate using a Multidrop 384 Reagent Dispenser (Thermo Fisher Scientific, Ermelo, The Netherlands). The plate was then read immediately for absorbance kinetically for 100 min at 595 nm at 25 °C using a Varioskan Flash Multimode Reader (Thermo Fisher Scientific, Ermelo, The Netherlands). The obtained results were normalized by dividing the slope measured in each well by the median of all slope values across the plate, and the processed coagulation chromatograms were plotted to visualize very fast coagulation, medium increased coagulation and anticoagulation, as previously described[37].

**In vivo experimentation**. All animal experiments were conducted using protocols approved by the Animal Welfare and Ethical Review Boards of the Liverpool School of Tropical Medicine and the University of Liverpool, and performed in specific pathogen-free conditions under licensed approval (PPL #4003718 and #P5846F90) of the UK Home Office and in accordance with the Animal [Scientific Procedures] Act 1986 and institutional guidance on animal care. All experimental animals (18–20 g [4–5 weeks old], male, CD-1 mice, Charles River, UK) were housed in groups of five with environmental enrichment, water and food ad libitum and their health monitored daily during acclimatization. The experimental design was based upon 3R-refined WHO-recommended protocols[28,41], with animals randomized and observers being blinded to the experimental condition. The median lethal doses (venom LD₅₀) used for E. ocellatus (Nigeria), E. carinatus (India), B. asper (Costa Rica), D. russelii (Sri Lanka) and B. arietans (Nigeria) venoms were previously determined[25,41,44,45].

**Preclinical efficacy via a preincubation model of envenoming**. For our initial in vivo experiments, we used 2.5 × the intravenous LD₅₀ doses of E. ocellatus (45 μg), E. carinatus (47.5 μg), B. asper (47 μg), D. russelii (20 μg) and B. arietans (54 μg) venoms in a 3R-refined version of the WHO-recommended[41] antivenom ED₅₀ neutralization experiment[25]. Groups of five mice received experimental doses that consisted of either: (a) venom only (2.5 × LD₅₀ dose); (b) venom and solo drug (60 μg); (c) solo drug-only (60 μg); (d) venom and a mix of two or three drugs (60 μg each); or (e) a mix of two or three drugs only (60 μg each). Drugs were dissolved in water, with the exception of varespladib, which was prepared as a 5 mg/ml stock in DMSO (2.5% in the final dose) due to solubility. All experimental doses were prepared to a volume of 200 μl in PBS and incubated at 37 °C for 30 min prior to their intravenous injection via the tail vein. Animals were monitored for 6 h, and euthanized via rising concentrations of CO₂ upon observation of previously defined humane end points that are predictors of lethality (e.g., seizure, pulmonary distress, paralysis, hemorrhage)[28]. Deaths, time of death, and survivors were recorded; where death/time of death actually represents the implementation of euthanasia based on defined humane end points.

**Preclinical efficacy via a 'challenge then treat' model of envenoming**. In these experiments, mice were challenged with venom intraperitoneally followed by delayed dosing of the marimastat and varespladib inhibitor mix 15 min later, as previously described[28]. For E. ocellatus, E. carinatus and B. arietans venoms we challenged mice with 5× i.v. LD₅₀ doses (90 μg, 95 and 108 μg, respectively), while higher doses were required to cause lethality with B. asper (303 μg, ~16× i.v. LD₅₀s) and D. russelii (13× i.v. LD₅₀, 105 μg) venoms in this model. All intraperitoneal venom doses consisted of a final volume of 100 μl in PBS. Drug doses were scaled up from 60 μg/mouse in the preincubation experiments outlined above to 120 μg/mouse here, in line with the (at least) doubling of the venom challenge dose from 2.5 × LD₅₀ to 5 × LD₅₀ (i.e., for E. ocellatus, E. carinatus and B. arietans). All inhibitor doses were delivered intraperitoneally 15 min after venom injection and consisted of 200 μl final volumes. The experimental groups comprised five mice receiving: (a) venom only + 200 μl PBS (15 min later); (b) venom + drug mix (120 μg marimastat and 120 μg varespladib, 15 min later); and (c) sham (100 μl PBS) + drug mix (15 min later). Experimental animals were monitored for 24 h, with humane end points for euthanasia, and data recording, performed as described above.

**Quantification of thrombin-antithrombin levels and thrombomodulin levels by ELISA**. For all experimental animals described above, blood samples were collected via cardiac puncture immediately post-euthanasia. Plasma was separated by centrifugation at $400 \times g$ for 10 min and stored at $-80\,°C$. We assessed the levels of thrombin-antithrombin complexes (TAT) and soluble thrombomodulin using mouse ELISA Kits (ab137994 and ab209880, Abcam), following the manufacturer's protocol. All available plasma samples (some were unobtainable via cardiac puncture due to extensive internal hemorrhage) were assessed if the time of death within the group varied, whereas three samples were randomly chosen if the time of death was the same (e.g., either very rapid death within 2 min, or survival until the end of the experiment [360 min or 24 h]). The resulting data was plotted as the median of duplicate measurements for each animal and is presented with standard deviations (SDs).

**Reporting Summary**. Further information on research design is available in the Nature Research Reporting Summary linked to this article.

## Data availability

The datasets generated and analyzed during the current study are available from the corresponding author on reasonable request. The raw data supporting the findings of this study and that are displayed in Figs. 2, 3, 4, 5, and 6, and Supplementary Figs. 2, 4, and 5, can be found in the Supplementary Data File (Supplementary Data S1). Data used to construct the species distributions displayed in Fig. 1 are freely available on the World Health Organization Venomous Snake Distribution database (https://apps.who.int/bloodproducts/snakeantivenoms/database/) and the IUCN Red List of Threatened Species database (https://www.iucnredlist.org). Source data are provided with this paper.

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

## Acknowledgements

We thank Paul Rowley for maintaining the snakes at the LSTM Herpetarium and for routine venom extractions, Mark Wilkinson for expertise relating to venom fractionation, and Joshua Longbottom for assistance generating the snake distribution maps. This study was supported by: (i) a Sir Henry Dale Fellowship to N.R.C. (200517/Z/16/Z) jointly funded by the Wellcome Trust and Royal Society, (ii) UK Medical Research Council (MRC) funded Research Grant (MR/S00016X/1) and Confidence in Concept Award (CiC19017) to R.A.H. and N.R.C., and (iii) a Wellcome Trust funded Biomedical Vacation Scholarship (207075/Z/17/) to R.S.

## Author contributions

L.-O.A., J.K and N.R.C. conceptualized the project; L.-O.A., C.A.D., R.S. and K.E.B. performed in vitro experiments; C.X. and J.K. performed venom nanofractionation experiments; S.A., J.A., E.C., C.A.D., R.A.H. and N.R.C. performed in vivo experiments. L.-O.A. performed data analyses. L.-O.A. and N.R.C. wrote the manuscript with input and approval from all other authors.

## Competing interests

The authors declare no competing interests.
