## [Peer Review File · Nature Communications]

Reviewers' Comments:

Reviewer #1:

Remarks to the Author:

The manuscript, A therapeutic combination of two small molecule toxin inhibitors provides pancontinental preclinical efficacy against viper snakebite, describes an exciting novel study, where the ability of a few well-selected small molecules to neutralize snake venom proteases and phospholipases are studied in a range of in vitro and in vivo experiments. Most interestingly, the authors demonstrate that by mixing some of the molecules, that the toxic effects of whole venom from vipers can be prevented/neutralized. The results are convincing and impactful and supported by validated and robust methods and will likely be of wide interest to researchers both within the field of snakebite envenoming and toxinology, but also beyond this field, given the increased awareness that snakebite envenoming has received in the last years.

When published, I find it likely that this manuscript will be a very influential paper in the field.

I recommend that the manuscript is accepted for publication, although perhaps some of the figures may need to be improved a bit in terms a readability (the fonts are a bit small in some of these).

I congratulate the authors for having performed such a nice study.

Andreas H. Laustsen

Reviewer #2:

Remarks to the Author:

The authors have undertaken a comprehensive clinical study of small molecules in snake envenoming that covers major viper species in the world. The experiments appear to be appropriate and undertaken well.

I have no major concerns, and the paper is written well. A couple of comments.

1. The discussion of use is a little confusing - it is suggested as a pre-clinical treatment (oral), which is reasonable. However, later it is suggested as the only treatment. One issue with venoms that have prolonged effects - eg. Echis, dosing for 2 weeks would be problematic, rather than simply giving antivenom after an initial oral dose. Perhaps discuss this.
2. Did the authors look at potential neurotoxic effects of *B. arietans* ? This would be important. Similarly for Russell's viper in which neurotoxicity is common in some countries.

Reviewer #3:

Remarks to the Author:

General comment

This impressive report describes a logical sequence of in vitro and in vivo mouse studies leading to the selection of a combination of two snake venom enzyme inhibitors that proved protective in mice against early lethal effects of the venoms of five medically important Viperidae from Africa, Asia and Latin America. Since the inhibitors, varespladib and marimastat, are orally bioavailable and have been licensed for human use in Phase II studies, they offer the possibility of becoming a first-aid, pre-hospital treatment for patients bitten by snakes in many countries, and they partially overcome the species-specificities and inadequacies of existing hyperimmune antiserum antivenoms.

The methods and results are clear and convincing, but I have three reservations about the paper as it is currently written.

- i. The imperfections and limitations of the two in vivo mouse models in reflecting the realities of human snakebite are insufficiently discussed and the authors' comments on dose translation are

too simplistic.

- ii. The practical difficulties of translating the results for human use are underestimated. To be suitable for distribution to snakebite-prone communities as a self-administered "snakebite pill" this enzyme combination would have to be acceptably safe among all ages, sizes, and co-morbidities of potential snakebite victims in a dosage that would be effective. Its use should not discourage the bitten person from urgently seeking medical care. This is a very tall order which should certainly not inhibit development of the idea but should moderate the level of optimism. Mass drug therapy has encountered problems in its application in a variety of diseases. Its attendant risks and practicalities must not be underestimated.
- iii. The language is unnecessarily immoderate (immodest), sounding unscientific in describing results which speak for themselves, and the prospects for translation of this work which are exciting and do not need to be overstated.

Detailed comments

Title

A therapeutic combination of two small molecule toxin inhibitors provides GEOGRAPHICALLY WIDE preclinical efficacy against viper snakebite IN RODENTS/ OR IN A RODENT MODEL

Abstract

Line 20 ?fiscal – In what sense? Do the authors mean financial?

23 Furthermore, using multiple in vivo mouse models of envenoming - Two models were used
25 prevents lethality caused by venom from the most medically-important vipers of Africa, South Asia and Central America. – In mice

26 Asia and Central America. Our findings strongly support – "Our findings support" the adjective is unnecessary

27 safe and affordable enzyme inhibitors – Surely "safe and affordable" for application for human pre-hospital use is highly speculative

Introduction

31 sub-Saharan Africa, South and Southeast Asia, and Central and South America. – Omission of Oceania

40 extensive interspecific [and intra-specific] variation in venom composition generic snakebite treatments – The term "generic treatment" has another meaning in pharmacy

53 intravenous delivery in a hospital setting – Antivenom is frequently given in health posts, clinics and dispensaries

67 these three enzymatic families.....are responsible for – Contribute to

72 Historically, small molecule toxin inhibitors have received limited attention – Since the word "historically" is used here and below, credit should be given to J-M Gutiérrez for the first studies of synthetic inhibitors such as batimastat (2000-). There is even earlier work on inhibitors of Crotalus atrox venom enzymes (JH Brown 1963)

84 despite the great promise – the adjective could be deleted

96 Our findings hold much promise for the future translation of combinations of small molecule toxin inhibitors into generic prehospital therapies for treating hemotoxic snakebites – This is a bold claim that should perhaps be more restrained by likely practicalities and limitations.

Results

138 all of the tested viper venoms displayed net procoagulant activity, with the exception of B. arietans, which had no effect on plasma clotting – B. arietans venom from East and South Africa has procoagulant activity

182 simultaneously inhibiting both procoagulant and anticoagulant venom toxins. – What about haemorrhagic activity which is potentially lethal and to which other venom toxins such as D. russelii Snake Venom Vascular Endothelial Growth Factors (VEGFs) may contribute?

206 via the tail vein, and is based on the gold standard – But most agree that existing in vivo models are woefully inadequate in representing human envenoming and therapeutic rescue

217 treatment with varespladib resulted in two early deaths.... – Really? Don't they mean "failed to prevent...."

228 Both toxin inhibitor mixtures resulted in full survival ... Delete the adjective "resulted in survival.." is sufficient

255 In a remarkable demonstration of the therapeutic potential of small molecule inhibitors –

Delete "remarkable"

281 is first administered intraperitoneally and then the test therapy is administered intraperitoneally – What was the rationale for choosing the ip route?

282 To this end, we injected venom from each of the five viper species in doses equivalent to at least 5 x the intravenous (iv) LD50 dose followed, 15 284 mins later, by a single dose of the inhibitor mixture – What was the basis for dose and timing?

296 combination resulted in extensive, prolonged [delete one of these adjectives], survival for at least 17 h after the venom-only controls suffered venom-induced lethality – In human snakebite envenoming, effects delayed long afterwards can be lethal (e.g. acute kidney injury)

297 Remarkably, all animals receiving... – This adverb could be deleted

312 B. asper, inhibiting toxins acting to disrupt the endothelium. – Only those associated with release of thrombomodulin; there are many other venom-targeted endothelial factors

Discussion

314-22 The first part of Discussion is unnecessarily repetitive with Introduction

329 These characteristics, combined with their oral formulation, provide an opportunity to provide prehospital treatment for snakebite – This is an exciting and important prospect, but some of the uncertainties about dosage, safety, and cost should be considered.

335 inducing SVMP and PLA2 toxin families, provides extensive preclinical protection against lethality... Why "extensive" protection?

358 which translates to 0.24 mg/kg and 0.48 mg/kg when applying animal to human dose conversion. – This dose conversion is highly questionable.

365-369 The safety of varespladib is a major concern since the dose necessary to produce protection in humans equivalent to that in the mouse model is completely unknown.

381 in these clinical studies is ~15-fold higher than the human equivalent dose used intraperitoneally in our animal model – this dose equivalence is highly speculative

385 Our in vivo venom neutralization studies convincingly demonstrate – The adjective could be deleted

394 of E. ocellatus venom when used as a solo therapy (Fig. 5A), strongly suggest – "suggest" is adequate

410 undoubtedly represents a highly promising future therapeutic – "represents a promising future therapeutic" seems adequate

424 Notwithstanding the clear and exciting snakebite therapeutic promise – "clear and exciting" could be deleted

426 preclinical studies are needed to explore the neutralizing efficacy of this combination in oral dosing regimens – A very important consideration

432 Finally, the considerable in vitro and in vivo venom neutralization data presented here – "considerable" could be deleted

437 varespladib, provide unparalleled pancontinental protection – This sounds overstated and is incorrect

439 While these findings hold much promise – "are promising" seems adequate

446 snake venoms, and thus strongly advocates – better without the adjective

447 generic, prehospital treatments to substantially reduce – "to reduce"

Responses to reviewer comments

Reviewer #1

The manuscript, A therapeutic combination of two small molecule toxin inhibitors provides pancontinental preclinical efficacy against viper snakebite, describes an exciting novel study, where the ability of a few well-selected small molecules to neutralize snake venom proteases and phospholipases are studied in a range of in vitro and in vivo experiments. Most interestingly, the authors demonstrate that by mixing some of the molecules, that the toxic effects of whole venom from vipers can be prevented/neutralized. The results are convincing and impactful and supported by validated and robust methods and will likely be of wide interest to researchers both within the field of snakebite envenoming and toxinology, but also beyond this field, given the increased awareness that snakebite envenoming has received in the last years. When published, I find it likely that this manuscript will be a very influential paper in the field.

I recommend that the manuscript is accepted for publication, although perhaps some of the figures may need to be improved a bit in terms a readability (the fonts are a bit small in some of these).

I congratulate the authors for having performed such a nice study.

We thank the reviewer for their review and positive words about our manuscript. In response to their suggestion, we have amended the size of some of the axis legends in a number of the figures to improve readability.

Reviewer #2

The authors have undertaken a comprehensive clinical study of small molecules in snake envenoming that covers major viper species in the world. The experiments appear to be appropriate and undertaken well.

I have no major concerns, and the paper is written well. A couple of comments.

We thank the reviewer for their time and careful consideration of our paper.

1. The discussion of use is a little confusing - it is suggested as a pre-clinical treatment (oral), which is reasonable. However, later it is suggested as the only treatment. One issue with venoms that have prolonged effects - eg. Echis, dosing for 2 weeks would be problematic, rather than simply giving antivenom after an initial oral dose. Perhaps discuss this.

We thank the reviewer for raising this valuable point. In response, we have added an additional sentence to the limitations part of the discussion section of the manuscript. The relevant text can be found below, with the new addition highlighted in yellow:

“Additional preclinical studies are needed to explore the neutralizing efficacy of this combination in oral dosing regimens, including repeat dosing experiments combined with pharmacokinetic analyses, to model the oral dose required to maintain effective concentrations of the drugs sufficient to provide prolonged protection from envenoming. This may be particularly challenging for cases where envenoming may result in prolonged treatment times, for example as the result of recurrence of coagulopathy or acute kidney injury, and thus additional model development addressing this point is needed.”

*2. Did the authors look at potential neurotoxic effects of *B. arietans* ? This would be important. Similarly for Russell's viper in which neurotoxicity is common in some countries.*

The predominant clinical signs of systemic envenoming by *B. arietans* are hypovolemic shock, haemorrhage, cardiotoxicity and myotoxicity rather than neurotoxicity. However, for Russell's viper venom, this is a relevant point. The *D. russelii* venom we used was from Sri Lanka, where clinical reports of mild neuromuscular paralysis have previously been described, and attributed to a presynaptic PLA₂ toxin (Silva et al. 2017. *Neurotoxicity Research* 31, 11-19), though other signs of systemic envenoming are more medically-important. We did not explicitly investigate neutralisation of neurotoxic PLA₂s in this study because: 1) our focus was on inhibiting venom-induced haemotoxicity and 2) neurotoxicity following envenoming is not observed in our animal (mouse) model when using this venom. However, other research, including from our group, has indicated that varespladib is capable of inhibiting PLA₂s (Xie et al. 2020. *Biomedicines* 8, e165), including presynaptic neurotoxins (Gutiérrez et al. 2020. *Toxins* 12, 131), and thus this would be a valuable study to pursue in the future.

Reviewer #3

This impressive report describes a logical sequence of in vitro and in vivo mouse studies leading to the selection of a combination of two snake venom enzyme inhibitors that proved protective in mice against early lethal effects of the venoms of five medically important Viperidae from Africa, Asia and Latin America. Since the inhibitors, varespladib and marimastat, are orally bioavailable and have been licensed for human use in Phase II studies, they offer the possibility of becoming a first-aid, pre-hospital treatment for patients bitten by snakes in many countries, and they partially overcome the species-specificities and inadequacies of existing hyperimmune antiserum antivenoms.

We thank the reviewer for their comprehensive review and detailed comments relating to the manuscript. Although there were a lot of relatively minor points to address, we found ourselves agreeing with the vast majority of the reviewer's thoughts, and consequently have implemented a number of changes to the manuscript. We thank the reviewer for going through the manuscript with such care, and enabling us to improve the paper.

The methods and results are clear and convincing, but I have three reservations about the paper as it is currently written.

i. The imperfections and limitations of the two in vivo mouse models in reflecting the realities

of human snakebite are insufficiently discussed and the authors' comments on dose translation are too simplistic.

The reviewer makes a sound point about the in vivo models – they do come with limitations, although the combination of the two models used herein remains the most robust way of assessing the preclinical efficacy of snakebite treatments at this time. To address the reviewer's point, we have added a sentence to the limitations section of the manuscript discussion, which now reads: *“Despite the combination of animal models used here providing confidence of broad anti-venom efficacy, these models remain limited in terms of accurately recapitulating cases of human envenoming (e.g. in terms of venom dose, route of venom delivery, treatment duration, etc). Thus, additional preclinical studies are needed to further explore...”*.

In terms of the reviewer's point regarding dose translation, these are necessarily facile (this word has now been inserted to explicitly emphasise this point) at this time, as there is no robust PK/PD-informed mouse bridging study data available to facilitate more robust predictions of human dosing. We plan to undertake this work in the future. The need for these future studies was already made in the limitations section of the manuscript discussion, but has now been further emphasised elsewhere in the text. In the discussion section on marimastat an additional sentence has been added which reads: *“However, murine bridging studies incorporating pharmacokinetic (PK) profiling coupled with pharmacodynamic (PD) assessments of venom neutralization are required in the future to enable accurate simulations of predicted human doses”*. In the varespladib section of the discussion the text *“though murine bridging study-based simulations of appropriate human doses are needed”* has been added. Finally, in the limitations section, some additional text has also been added and this part on dosing now reads *“potential drug-drug interactions at PK/PD-informed human doses also need to be robustly assessed”*.

ii. The practical difficulties of translating the results for human use are underestimated. To be suitable for distribution to snakebite-prone communities as a self-administered “snakebite pill” this enzyme combination would have to be acceptably safe among all ages, sizes, and co-morbidities of potential snakebite victims in a dosage that would be effective. Its use should not discourage the bitten person from urgently seeking medical care. This is a very tall order which should certainly not inhibit development of the idea but should moderate the level of optimism. Mass drug therapy has encountered problems in its application in a variety of diseases. Its attendant risks and practicalities must not be underestimated.

This is a fair point from the reviewer, although perhaps overstated in the context of this ‘preclinical testing’ paper. We have toned down some of the text relating to our ‘optimism’ (see responses to the various points below, for example) and added an additional sentence to the limitations section of the discussion stating that challenges remain associated with translation and health seeking behaviour. Clearly, there are many challenges to overcome between promising preclinical research and the delivery of an effective and successfully used clinical product. But as the reviewer states, these challenges should not inhibit development of the idea. The new sentence reads as follows: *“Finally, the successful delivery and uptake of any prehospital snakebite treatment comes with a number of implementation challenges, including ensuring (i) acceptable safety profiles across the target population (e.g. both children and adults) and (ii) that health seeking behaviour after initial treatment is strongly*

promoted so that patients are carefully monitored in case additional (i.e. doses) or complementary (i.e. antivenom) treatment is required”.

iii. The language is unnecessarily immoderate (immodest), sounding unscientific in describing results which speak for themselves, and the prospects for translation of this work which are exciting and do not need to be overstated.

We have modified the manuscript in response to a number of the reviewer’s comments relating to this point (see outlined in detail below).

Detailed comments

Title

A therapeutic combination of two small molecule toxin inhibitors provides GEOGRAPHICALLY WIDE preclinical efficacy against viper snakebite IN RODENTS/ OR IN A RODENT MODEL

We have modified the title slightly in response to this comment (switched “pancontinental” to “broad”), but have insufficient words available to us due to journal requirements to add additional text. To compensate, the word “murine” has been added to the abstract twice to make it clear to the reader that the preclinical models used mice. This was a good spot by the reviewer though, because it was an oversight that the abstract did not contain this information already.

Line 20 ?fiscal – In what sense? Do the authors mean financial?

Amended to “financial”

23 Furthermore, using multiple in vivo mouse models of envenoming - Two models were used

Amended to “using murine *in vivo* models of envenoming”

25 prevents lethality caused by venom from the most medically-important vipers of Africa, South Asia and Central America. – In mice

Amended to “prevents murine lethality caused by venom from”

26 Asia and Central America. Our findings strongly support – “Our findings support” the adjective is unnecessary

Changed as suggested.

27 safe and affordable enzyme inhibitors – Surely “safe and affordable” for application for human pre-hospital use is highly speculative

Changed to “translation of combinations of repurposed small molecule-based toxin inhibitors as broad-spectrum therapeutics for snakebite”.

Introduction

31 sub-Saharan Africa, South and Southeast Asia, and Central and South America. – Omission of Oceania

Corrected.

40 extensive interspecific [and intra-specific] variation in venom composition generic snakebite treatments – The term “generic treatment” has another meaning in pharmacy

Our meaning is now further clarified via the addition of “(i.e. pancontinental)”.

53 intravenous delivery in a hospital setting – Antivenom is frequently given in health posts, clinics and dispensaries

Thanks for spotting this error. Amended to “healthcare facility”.

67 these three enzymatic families.....are responsible for – Contribute to

We deem ‘contribute to’ as insufficiently strong, but have amended the text to “largely responsible for”

*72 Historically, small molecule toxin inhibitors have received limited attention – Since the word "historically" is used here and below, credit should be given to J-M Gutiérrez for the first studies of synthetic inhibitors such as batimastat (2000-). There is even earlier work on inhibitors of *Crotalus atrox* venom enzymes (JH Brown 1963)*

Unfortunately, we are limited by journal restrictions as to the number of references we can provide here. We have cited the work of J-M Gutierrez on batimastat and marimastat though – using the Arias et al. reference in preference of the one suggested by the reviewer. We have removed the word “historically” to avoid the suggestion that our review of the literature includes every paper on this topic.

84 despite the great promise – the adjective could be deleted

Amended as suggested.

96 *Our findings hold much promise for the future translation of combinations of small molecule toxin inhibitors into generic prehospital therapies for treating hemotoxic snakebites – This is a bold claim that should perhaps be more restrained by likely practicalities and limitations.*

We agree that this statement is too strong and overstates the case. We have amended the text to more accurately reflect our findings. It now reads: “*Our findings suggest that combinations of small molecule toxin inhibitors are promising drug leads for the future development of generic prehospital therapies for treating hemotoxic snakebites.*”

Results

138 *all of the tested viper venoms displayed net procoagulant activity, with the exception of B. arietans, which had no effect on plasma clotting – B. arietans venom from East and South Africa has procoagulant activity*

As stated in the manuscript, our *B. arietans* venom was from Nigeria, and it clearly has no procoagulant activity in our plasma coagulation assay.

182 *simultaneously inhibiting both procoagulant and anticoagulant venom toxins. – What about haemorrhagic activity which is potentially lethal and to which other venom toxins such as D. russelii Snake Venom Vascular Endothelial Growth Factors (VEGFs) may contribute?*

For obvious reasons, it is problematic to test in vitro inhibition of every potential venom activity. Indeed, part of the point of this study is to assess how few therapeutic molecules one can include in a combination to treat snakebite. If we wished to inhibit every toxin family across all snakes, we would end up with many distinct therapeutic molecules that would make translation extremely impractical. Thus, we rationally investigated inhibitors against the key viper toxin families found abundantly across many different snake species, as clearly outlined in the introduction. Generally speaking, SVMs are known to be the major contributors to viperid haemorrhage.

206 *via the tail vein, and is based on the gold standard – But most agree that existing in vivo models are woefully inadequate in representing human envenoming and therapeutic rescue*

Please refer to our response to this point above.

217 *treatment with varespladib resulted in two early deaths.... – Really? Don't they mean “failed to prevent....”*

Good point. We have modified the text to now read “treatment with varespladib failed to prevent lethality over the experimental time course, with two early deaths and three later deaths observed”.

228 *Both toxin inhibitor mixtures resulted in full survival ... Delete the adjective “resulted in survival..” is sufficient*

Amended as suggested.

255 *In a remarkable demonstration of the therapeutic potential of small molecule inhibitors – Delete “remarkable”*

Superlative removed.

281 *is first administered intraperitoneally and then the test therapy is administered intraperitoneally – What was the rationale for choosing the ip route?*

The intraperitoneal route was used to ensure an acceptable time window to measure venom neutralization. Mice dosed intravenously rapidly succumb to envenoming (often within the 15 mins window between challenge and treatment), thus an alternative route was required. We used the ip route following our previous successful application of this model in a paper cited in the manuscript methods section (Albulescu et al. 2020 *Science Translational Medicine*), which further explains this model.

282 *To this end, we injected venom from each of the five viper species in doses equivalent to at least 5 x the intravenous (iv) LD50 dose followed, 15 284 mins later, by a single dose of the inhibitor mixture – What was the basis for dose and timing?*

The venom dose was selected to ensure a monitoring window of at least 17 h (ideally 20 h) post-venom only control lethality. I.e. venom-only treated animals should succumb within 7 h to ensure 17 h of monitoring of lethality in the presence of drug. This is detailed in the results section of the manuscript: “*to ensure mortality occurred within 7 h, thus leaving a 17 h window for measuring prolonged survival in the treatment groups*”. The timing of treatment 15 mins later follows our previous use of this model to enable comparisons; this is described in the methods: “*In these experiments, mice were challenged with venom intraperitoneally followed by delayed dosing of the marimastat and varespladib inhibitor mix 15 mins later, as previously described²⁸*”. The drug dose selected was a facile scale up from the preincubation experiments – again detailed in the methods – “*drug doses were scaled up from 60 µg/mouse in the preincubation experiments outlined above to 120 µg/mouse here, in line with the (at least) doubling of the venom challenge dose from 2.5 ×LD₅₀ to 5 ×LD₅₀ (i.e. for E. ocellatus, E. carinatus and B. arietans).*”

296 *combination resulted in extensive, prolonged [delete one of these adjectives], survival for at least 17 h after the venom-only controls suffered venom-induced lethality – In human snakebite envenoming, effects delayed long afterwards can be lethal (e.g. acute kidney injury)*

This is a reasonable point, but we cannot address this with our murine model. Our ethical approvals permits a maximum duration of 24 h monitoring during these severe-rated procedures. The point about the murine model not completely recapitulating the effects of

human envenoming has also been made above (see earlier responses), and we have added additional text to the limitations section of the manuscript to highlight this point, including specifically mentioning the potential challenge for prolonged treatment times (listing acute kidney injury and coagulopathy). Also, “extensive” has been removed, as requested.

297 Remarkably, all animals receiving.... – This adverb could be deleted

Amended as suggested.

312 B. asper, inhibiting toxins acting to disrupt the endothelium. – Only those associated with release of thrombomodulin; there are many other venom-targeted endothelial factors

We agree. The text has been modified now to read “inhibiting toxins acting to disrupt certain components of the endothelium”.

Discussion

314-22 The first part of Discussion is unnecessarily repetitive with Introduction

We prefer to keep this text to remind the reader and further emphasise the rationale of the study since the journal is multi-disciplinary.

329 These characteristics, combined with their oral formulation, provide an opportunity to provide prehospital treatment for snakebite – This is an exciting and important prospect, but some of the uncertainties about dosage, safety, and cost should be considered.

Here we are talking generically about small molecule toxin inhibitors, so adding text on dosage, safety and cost would be out of place, since this depends on the molecule. We have adjusted the text to temper it slightly and it now reads “provide an opportunity to explore their utility as prehospital treatments”. Specific points about dosage and safety of marimastat and varespladib are discussed later on in the discussion.

335 inducing SVMP and PLA2 toxin families, provides extensive preclinical protection against lethality.... Why “extensive” protection?

Good spot - we have removed this imprecise wording.

358 which translates to 0.24 mg/kg and 0.48 mg/kg when applying animal to human dose conversion. – This dose conversion is highly questionable.

Please see our response above to this point.

365-369 The safety of varespladib is a major concern since the dose necessary to produce protection in humans equivalent to that in the mouse model is completely unknown.

One could argue that this is true of all (repurposed) drugs. We have added some extra text to this section highlighting again the need for bridging studies in the future: “*though bridging study-based simulations of appropriate human doses are needed*”.

381 in these clinical studies is ~15-fold higher than the human equivalent dose used intraperitoneally in our animal model – this dose equivalence is highly speculative

See earlier responses.

385 Our in vivo venom neutralization studies convincingly demonstrate – The adjective could be deleted

Amended as suggested.

394 of E. ocellatus venom when used as a solo therapy (Fig. 5A), strongly suggest – “suggest” is adequate

Amended as suggested.

410 undoubtedly represents a highly promising future therapeutic – “represents a promising future therapeutic” seems adequate

Amended to “DMPS remains a promising”.

424 Notwithstanding the clear and exciting snakebite therapeutic promise – “clear and exciting” could be deleted

Amended to “notwithstanding the apparent therapeutic promise of this small molecule toxin inhibitor combination for treating snakebite,”

426 preclinical studies are needed to explore the neutralizing efficacy of this combination in oral dosing regimens – A very important consideration

Indeed.

432 Finally, the considerable in vitro and in vivo venom neutralization data presented here – “considerable” could be deleted

Agreed – amended as suggested.

437 varespladib, provide unparalleled pancontinental protection –This sounds overstated and is incorrect

We agree that wording is imprecise – we were trying (perhaps poorly) to convey the fact that this drug combination neutralises venoms from snakes on different continents. Reworded to “provide broad protection against...”

439 While these findings hold much promise – “are promising” seems adequate

We prefer this as it is.

446 snake venoms, and thus strongly advocates – better without the adjective

Agreed – amended as suggested.

447 generic, prehospital treatments to substantially reduce – “to reduce”

Amended as suggested.

Reviewers' Comments:

Reviewer #2:

Remarks to the Author:

The authors have responded to my comments, and also the manuscript is improved with the modifications based on reviewer 3

Reviewer #3:

Remarks to the Author:

I find the authors' responses to my review entirely satisfactory.